# Caliciviral protein-based artificial translational activator for mammalian gene circuits with RNA-only delivery

Hideyuki Nakanishi [1,2] & Hirohide Saito [1✉]

Synthetic RNA-based gene circuits enable sophisticated gene regulation without the risk of insertional mutagenesis. While various RNA binding proteins have been used for translational repression in gene circuits, the direct translational activation of synthetic mRNAs has not been achieved. Here we develop Caliciviral VPg-based Translational activator (CaVT), which activates the translation of synthetic mRNAs without the canonical 5'-cap. The level of translation can be modulated by changing the locations, sequences, and modified nucleosides of CaVT-binding motifs in the target mRNAs, enabling the simultaneous translational activation and repression of different mRNAs with RNA-only delivery. We demonstrate the efficient regulation of apoptosis and genome editing by tuning translation levels with CaVT. In addition, we design programmable CaVT that responds to endogenous microRNAs or small molecules, achieving both cell-state-specific and conditional translational activation from synthetic mRNAs. CaVT will become an important tool in synthetic biology for both biological studies and future therapeutic applications.

[1] Department of Life Science Frontiers, Center for iPS Cell Research and Application, Kyoto University, 53 Kawahara-Cho, Shogoin, Sakyo-Ku, Kyoto 606-8507, Japan. [2] Present address: Department of Biofunction Research, Institute of Biomaterials and Bioengineering, Tokyo Medical and Dental University (TMDU), 2-3-10 Kanda-Surugadai, Chiyoda-ku, Tokyo 101-0062, Japan. ✉email: hirohide.saito@cira.kyoto-u.ac.jp

Mammalian synthetic biology provides a promising tool for both biological studies and medical applications, as it enables the sophisticated regulation of endogenous and exogenous gene expressions. Various proteins (e.g., transcription factors, receptors, and apoptotic proteins) have been utilized as components of mammalian synthetic biology[1]. Currently, most of these components are delivered to cells as a DNA format, but DNA format risks insertional mutagenesis, which may cause serious problems in therapeutic applications. As an alternative, synthetic messenger RNAs (mRNAs) are promising vectors because they do not cause insertional mutagenesis[2]. In addition, synthetic mRNAs containing modified nucleosides can avoid nucleic acid-mediated induction of inflammation[3–5].

Although synthetic mRNAs are useful for mammalian synthetic biology, the available regulatory components for mRNA-based gene circuits (RNA circuits) are much less than those for DNA-based gene circuits (DNA circuits). Especially, while transcriptional activators are important components in DNA circuits[6–8], the direct activation of gene expression is difficult in RNA circuits. Rather, in current RNA circuits, the activation of gene expressions has been achieved by the repression of translational repressors by combining multiple RNA-binding proteins (RBPs)[9,10,11]. However, in such indirect methods, the number of necessary components tend to increase, complicating the system.

In this study, we develop Caliciviral VPg-based Translational activator (CaVT), which enables the direct translational activation of synthetic mRNAs in RNA circuits (Fig. 1). CaVT is composed of MS2 coat protein (MS2CP), which is a motif-specific RBP, and a caliciviral VPg protein, which acts as a substitute 5′-cap structure[12]. CaVT binds to its target RNA motif in the 5′ UTRs of mRNAs without a canonical 5′-cap to directly activate their translation (Fig. 2a). The translational activation level can be modulated by changing the locations, sequences, and modified nucleosides of the target motif, and even translational repression can be achieved. This characteristic of CaVT enables different regulations of multiple mRNAs by a single protein (i.e., while one mRNA is translationally activated, another mRNA is translationally repressed through the RNA–protein interaction). Indeed, we simultaneously activate and repress the translation of proapoptotic and antiapoptotic proteins using bifunctional CaVT, making it possible to perform efficient cell-fate regulation with RNA-only delivery. CaVT-based RNA circuits efficiently regulate genome editing by the activation and repression of Cas9 and anti-CRISPR AcrIIA4 translation, respectively. In addition, we develop microRNA (miRNA)-responsive CaVT-based RNA circuits, in which the translation of two mRNAs are simultaneously upregulated and downregulated by one miRNA, by inserting the target site of the miRNAs into mRNAs that express CaVT. This miRNA-

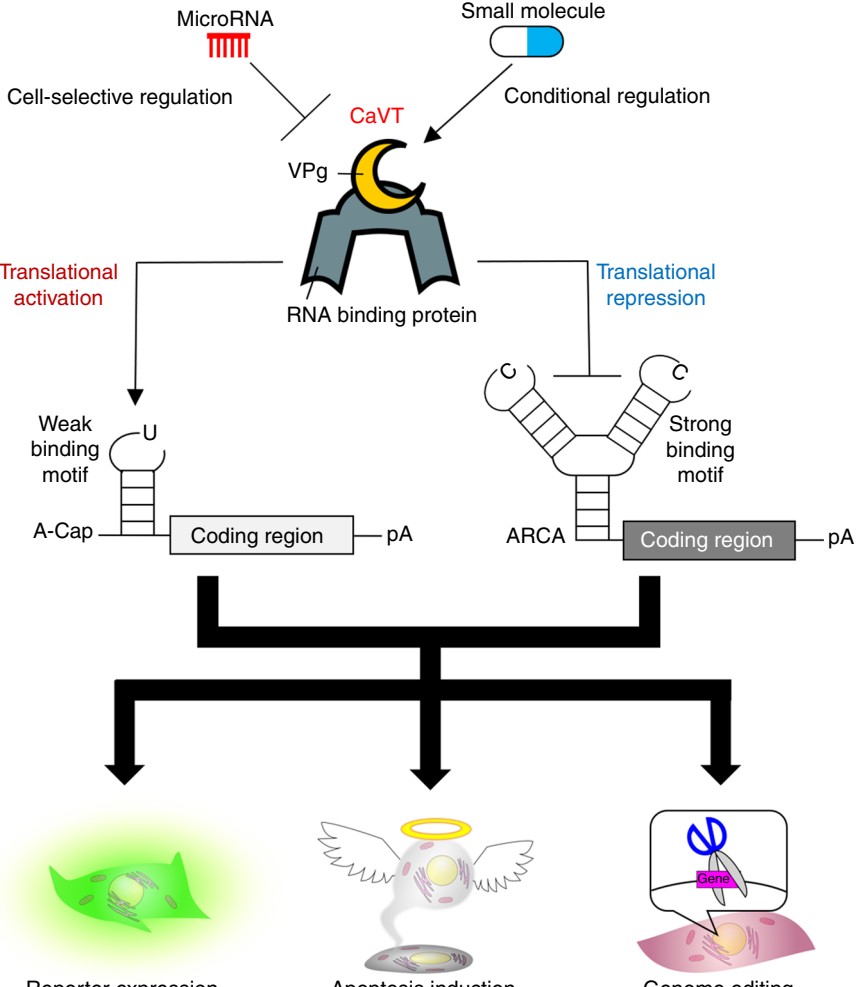

**Fig. 1 Graphical abstract of translational regulation by CaVT.** CaVT can act as both a translational activator and repressor. While mRNAs containing a weak binding motif are translationally activated by CaVT, mRNAs containing a strong binding motif are translationally repressed by CaVT. Through these translational regulations, CaVT can be used to control reporter expression, apoptosis induction, and genome editing. The implementation of microRNA- or small molecule responsiveness to CaVT enables cell selective or conditional regulation of these biological phenomena.

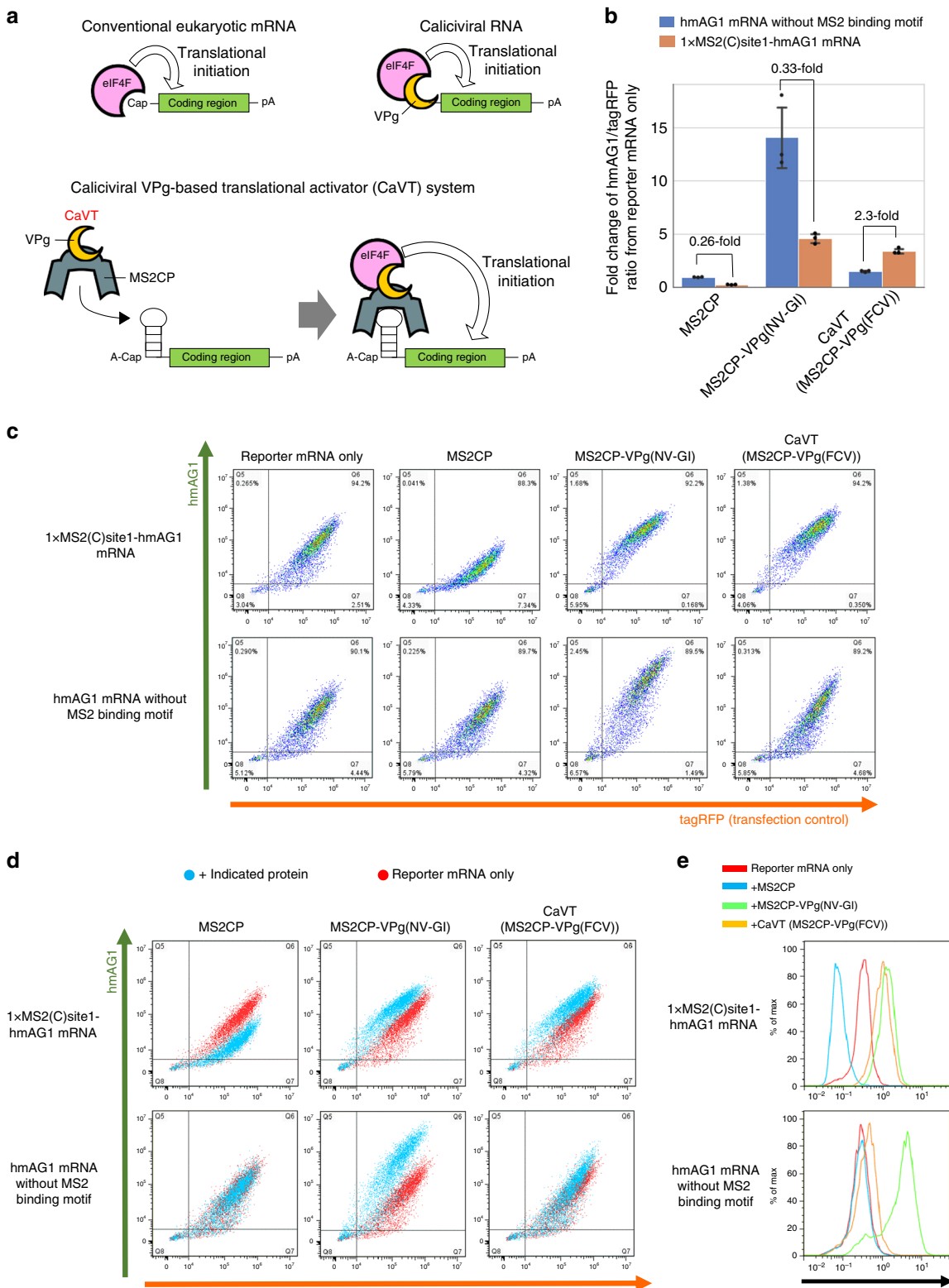

responsive RNA circuit enables the cell-selective regulation of genome editing. Finally, we achieve small molecule drug-inducible translational activation by combining CaVT and drug-inducible hetero-dimerization domains, in which the translational activation level can be modulated by changing the concentration of the drug. CaVT provides a promising tool for mammalian synthetic biology and will help with the design of more user-friendly, cell-type-specific, and conditional RNA circuits.

## Results

**Fusion of MS2CP and VPg to develop a translational activator**. In the natural life cycle of calicivirus, VPg proteins covalently bind with the 5'-ends of genomic and subgenomic RNAs[12]. Because the covalent bond-formation is associated with transcription by the caliciviral RNA-dependent RNA polymerase[13], it is difficult to bind VPg proteins with synthetic mRNAs in the same manner. Thus, we utilized a dlFG variant of MS2CP[14,15],

**Fig. 2 Comparison of VPg(FCV) and VPg(NV-GI) to construct the translational activator for synthetic mRNAs.** HeLa cells were co-transfected with hmAG1 mRNAs with or without MS2-binding motif (cap analog: A-cap), tagRFP mRNA, and the mRNAs of the indicated proteins. Details of the transfection conditions are described in the Supplementary Methods. The fluorescence was measured by flow cytometry. **a** Schematic diagrams of the mechanisms of translational initiation. For canonical eukaryotic mRNAs, eIF4F recognizes 5′-cap structures (top left), while for caliciviral RNAs, VPg proteins act as substitutes of the 5′-cap structures (top right). In the case of Caliciviral VPg-based Translational activator (CaVT), MS2CP binds to its target motif at the 5′ UTRs of synthetic mRNAs. eIF4F recognizes the VPg and initiates translation. The basal translation level of these synthetic mRNAs is low because A-cap, a translationally inactive cap analog, is fused with the 5′-end instead of a canonical cap (bottom). **b** Fold change of the hmAG1/tagRFP ratio caused by each indicated protein. Means of the hmAG1/tagRFP ratio were normalized by the hmAG1/tagRFP ratio in the reporter mRNA only sample. The bar graph shows the average of three independent experiments (mean ± SD). Source data are provided as a Source Data file. **c** Representative two-dimensional dot plots of hmAG1 and tagRFP. **d** Superimposition of the dot plots shown in (**c**). Cells transfected with mRNA to express the indicated proteins are shown as cyan, while cells transfected with only reporter mRNAs are shown as red. **e** Representative histograms of the hmAG1/tagRFP ratio in cells expressing both hmAG1 and tagRFP.

which is a motif-specific RBP, for the interaction between VPg and synthetic mRNAs without canonical 5′-cap structures (Fig. 2a). To investigate whether the MS2CP-mediated binding of VPg can activate the translation of mRNAs, we prepared the mRNAs that express MS2CP fused to norovirus GI (NV-GI)- or feline calicivirus (FCV)-derived VPg proteins (MS2CP-VPg(NV-GI) and MS2CP-VPg(FCV), respectively). As the target of these proteins, human codon-optimized monomeric Azami Green (hmAG1)-coding mRNA containing the binding motif of MS2CP in its 5′ end (1xMS2(C)site1-hmAG1 mRNA) was also prepared. These mRNAs and a transfection control tagRFP mRNA were co-transfected into HeLa cells, and the hmAG1/tagRFP ratio was analyzed to monitor translational activation. Although both MS2CP-VPg(NV-GI) and MS2CP-VPg(FCV) activated hmAG1 translation, MS2CP-VPg(NV-GI) showed high nonspecific translational activation of mRNA without the MS2-binding motif, and MS2CP-mediated binding to the target mRNA decreased its activation level (Fig. 2b–e). In contrast, MS2CP-VPg(FCV) showed low nonspecific translational activation, and the addition of the MS2-binding motif to hmAG1 mRNA resulted in 2.3-fold increase of MS2CP-VPg(FCV)-mediated translational activation (Fig. 2b–e). In contrast, MS2CP or unfused VPg(FCV) protein alone did not show a MS2-binding motif-dependent increase of translational activation (Fig. 2b–e; Supplementary Fig. 1). From these results, we concluded that MS2CP-VPg(FCV) is more suitable for the translational activation of specific target mRNAs. We named this fusion protein Caliciviral VPg-based Translational activator (CaVT) and used it in the following experiments.

To further confirm that the CaVT-mediated increase of translation was caused by a cap-mimicking effect of VPg, we investigated the effect of CaVT on 1xMS2(C)site1-hmAG1 mRNA already capped with translationally active cap analog (Anti-Reverse Cap Analog; ARCA). In the case of ARCA-capped 1xMS2(C)site1-hmAG1 mRNA, the binding of CaVT induced translational repression rather than translational activation (Supplementary Fig. 2), which was a similar effect as VPg-unfused MS2CP[16].

**Optimization of target mRNAs for CaVT.** To investigate the relationship between affinity and translational activation, we next used two MS2CP variants: conventional MS2CP and its V29I mutant, which has high affinity for the MS2-binding motif[15]. We also designed six different types of hmAG1 mRNA variants (Fig. 3a). These hmAG1 mRNAs contained one of the two MS2-binding motif variants (C or U), in which the C variant has a higher affinity for MS2CP than the U variant in a native nucleoside context[17]. The locations of these motifs were 5′ end (site1), the center of 5′ UTR (site2), or immediately upstream of the Kozak sequence (site3). In addition, we prepared these hmAG1 mRNA variants to have two different nucleoside compositions: N1-methyl-pseudouridine (N1mΨ) instead of uridine,

and pseudouridine (Ψ) and 5-methylcytidine (5mC) instead of uridine and cytidine, respectively. Our recent study suggested that MS2CP could bind to N1mΨ-containing RNAs more strongly than to Ψ/5mC-containing RNAs[18]. These CaVT and hmAG1 mRNA variants were co-transfected into HeLa cells, and the hmAG1 fluorescence was measured by using a flow cytometer (Fig. 3b, c, and Supplementary Fig. 3). The higher translational activation was achieved by the U-variant motif containing N1mΨ and the C-variant motif containing Ψ/5mC, and the most suitable location of the motif was site2 (7.1- to 8.2-fold increase of translational activation by CaVT) (Fig. 3b, c, and Supplementary Fig. 3). Notably, an increase of the CaVT/hmAG1 mRNA ratio from one-ninth to one-fourth did not improve the translational activation level (Supplementary Figs. 4 and 5).

The highest affinity combination (MS2CP(V29I) and N1mΨ-containing mRNA with a C-variant motif) showed the lowest translational activation (Fig. 3b and Supplementary Fig. 4a). Based on these results, we hypothesized that exceeding optimal affinity may decrease the translational activation level. To examine this hypothesis, we designed hmAG1 mRNA containing two copies of the C-variant motif stabilized by a scaffold (2xScMS2(C)-hmAG1), which has even higher affinity than the conventional C-variant motif (Fig. 4a)[10]. Interestingly, the translation of 2xScMS2(C)-hmAG1 mRNA (containing N1mΨ) was not activated but repressed by CaVT (Fig. 4b–d, Supplementary Figs. 6a and 7). The translational repression by CaVT was more evident when we co-transfected ARCA-capped 2xScMS2(C)-hmAG1 mRNA (Fig. 4e and Supplementary Fig. 6b). These results indicate that we can modulate the translational level in opposing directions (activation and repression) by engineering locations of the motif and the affinity between CaVT and mRNA. Because efficient translational activation and repression were observed in mRNAs containing N1mΨ (Figs. 3–4, and Supplementary Figs. 3–7), N1mΨ-containing mRNAs were used in the following experiments.

**Cell-fate regulation by CaVT.** The killing of undesired cells by regulating apoptotic protein expression has important applications in synthetic biology[19]. Cell-killing gene circuits could be used for cancer gene therapies or cell therapies[20,21]. However, sophisticated apoptosis regulation has remained a challenge due to leakiness of the protein expression and the circuit complexity. Thus, we designed an efficient apoptosis-regulatory system using CaVT-mediated, translational activation and repression of apoptotic proteins. First, we prepared mRNA (without canonical 5′-cap) that contains human Bax (pro-apoptotic protein) gene and a U-variant motif in the middle of its 5′ UTR (1xMS2(U) site2-Bax) (Fig. 5a, left). Co-transfection of CaVT mRNA increased the number of apoptotic cells by 1xMS2(U)site2-Bax, but even in the absence of CaVT, some portion of apoptotic induction was observed (Fig. 5b–d).

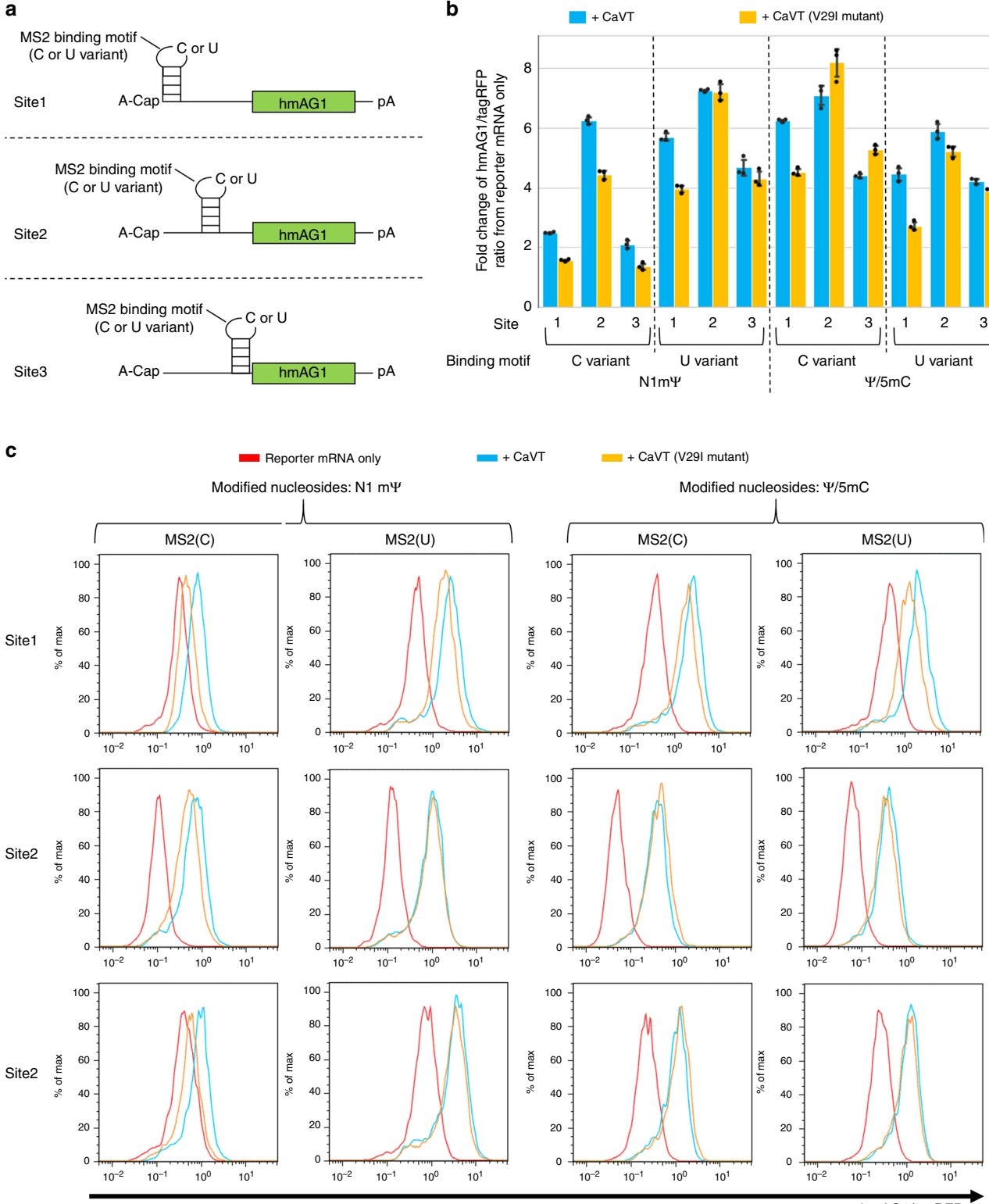

**Fig. 3 Effects of modified nucleosides, sites, and variants of MS2-binding motif and of variants of MS2CP on translational activation by CaVT.** HeLa cells were co-transfected with hmAG1 mRNAs with MS2-binding motif (cap analog: A-cap, modified nucleosides: N1mΨ or Ψ/5mC), tagRFP mRNA, and an mRNA that expresses CaVT or its V29I mutant. The fluorescence was measured by a flow cytometer. **a** Schematic diagrams of hmAG1 mRNAs with MS2 binding motif. **b** CaVT (or its V29I mutant)-mediated fold change of the hmAG1/tagRFP ratio in cells transfected with the indicated reporter mRNAs. Means of the hmAG1/tagRFP ratio in each cell expressing both hmAG1 and tagRFP were calculated and normalized by the hmAG1/tagRFP ratio in reporter mRNA only samples. The bar graph shows the average of three independent experiments (mean ± SD). Source data are provided as a Source Data file. **c** Representative histograms of the hmAG1/tagRFP ratio in cells expressing both hmAG1 and tagRFP. While cells transfected with only reporter mRNAs are shown as red, cells transfected with mRNA that express CaVT or its V29I mutant are shown as cyan and orange, respectively.

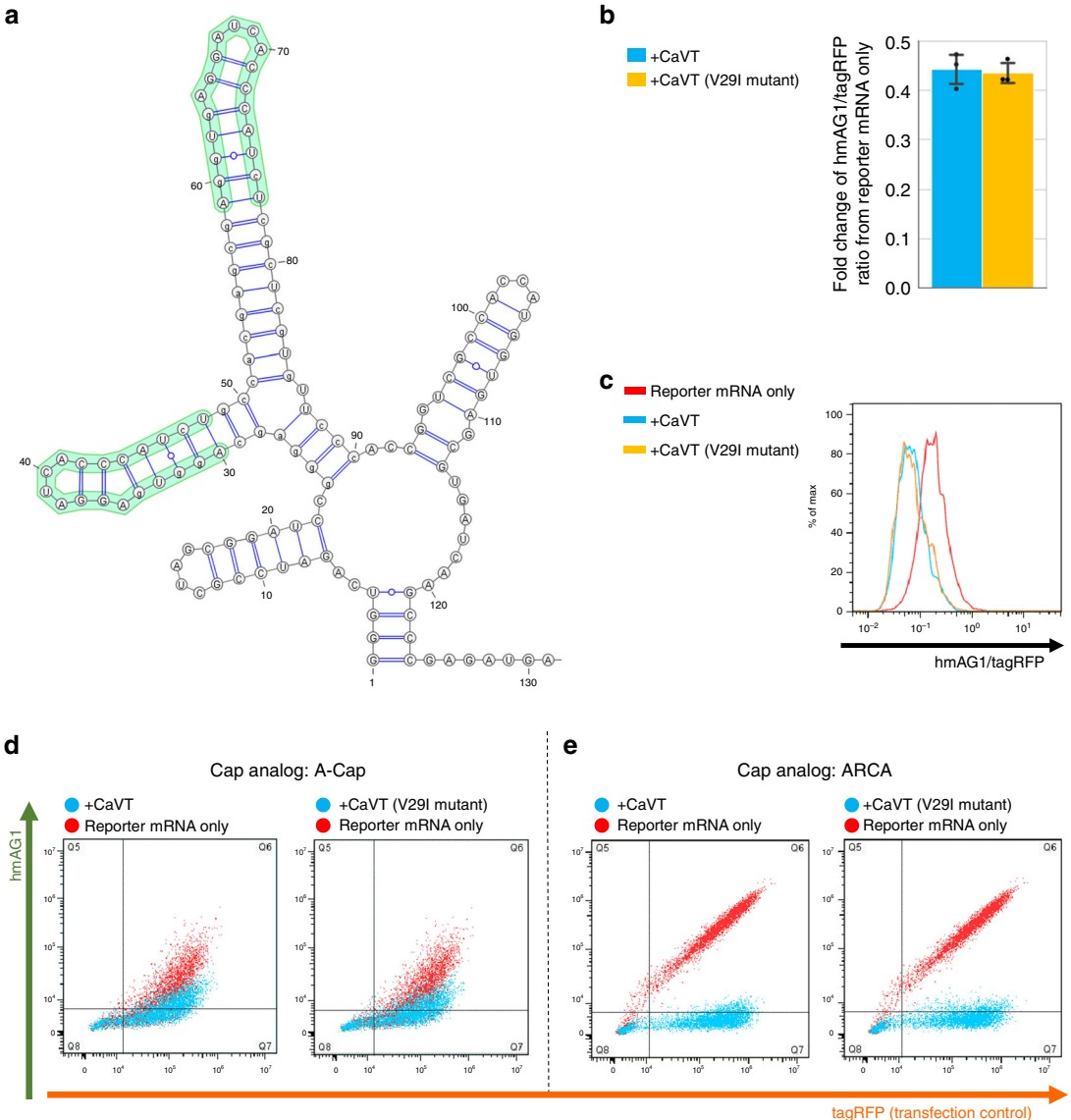

**Fig. 4 CaVT-mediated translational repression of the mRNA containing the strong binding motif. a** The predicted RNA secondary structure of MS2 binding motifs in the 5′ UTR of 2xScMS2(C)-hmAG1 mRNA. The structure was predicted by using ParasoR[54] and visualized by using VARNA[55]. **b** CaVT (or its V29I mutant)-mediated fold change of the hmAG1/tagRFP ratio in cells transfected with the indicated reporter mRNAs. Means of the hmAG1/tagRFP ratio in each cell expressing both hmAG1 and tagRFP were calculated and normalized by the hmAG1/tagRFP ratio in reporter mRNA-only samples. The bar graph shows the average of three independent experiments (mean ± SD). Source data are provided as a Source Data file. **c** Representative histograms of the hmAG1/tagRFP ratio in cells expressing both hmAG1 and tagRFP. Cells transfected with only reporter mRNAs are shown as red, with mRNA that express CaVT as cyan, and with mRNA that express the CaVT V29I mutant as orange. **d, e** Representative superimposed dot plots of cells transfected with 2xScMS2(C)-hmAG1 mRNA (A-capped, 360 ng/well (**d**) or ARCA-capped, 20 ng/well (**e**)). Cells co-transfected with mRNA that express CaVT or its V29I mutant are shown as cyan, and cells transfected with only reporter mRNAs are shown as red.

When we transfected 1xMS2(U)site2-hmAG1, some leaky expression was observed in the absence of CaVT (Supplementary Figs. 3 and 5). Based on the results of the hmAG1 experiments, we considered the leaky expression of Bax may be the cause of apoptosis in the absence of CaVT. To reduce the apoptotic effect caused by this leaky expression, we next designed mRNA coding an antiapoptotic protein, Bcl-xL[22], which directly binds with Bax and inhibits apoptosis. The Bcl-xL mRNA, named 2xScMS2(C)-BclxL, contains two copies of the C variant motif stabilized by the scaffold, which should cause CaVT-mediated translational repression of the flanking coding region. Thus, CaVT should simultaneously activate and repress the translation of 1xMS2(U) site2-Bax and 2xScMS2(C)-BclxL, respectively (Fig. 5a, right). In the absence of CaVT, the co-transfection of 1xMS2(U)site2-Bax

and 2xScMS2(C)-BclxL showed no increase of apoptotic cells compared with mRNA-untreated cells. In contrast, the additional co-transfection of CaVT mRNA significantly increased the number of apoptotic cells (Fig. 5b–d). These results indicate that our CaVT-mediated translational regulation system enables sophisticated cell-fate regulation by the simultaneous activation and repression of different mRNAs by a single protein.

**CaVT-mediated regulation of genome editing.** Next, we aimed to control genome editing with CaVT (Fig. 6a). We first prepared mRNA for the translational activation of *Streptococcus pyogenes*-derived Cas9 (1xMS2(U)site2-SpCas9). Cas9 is a target-programmable nuclease that makes a complex with a guide RNA

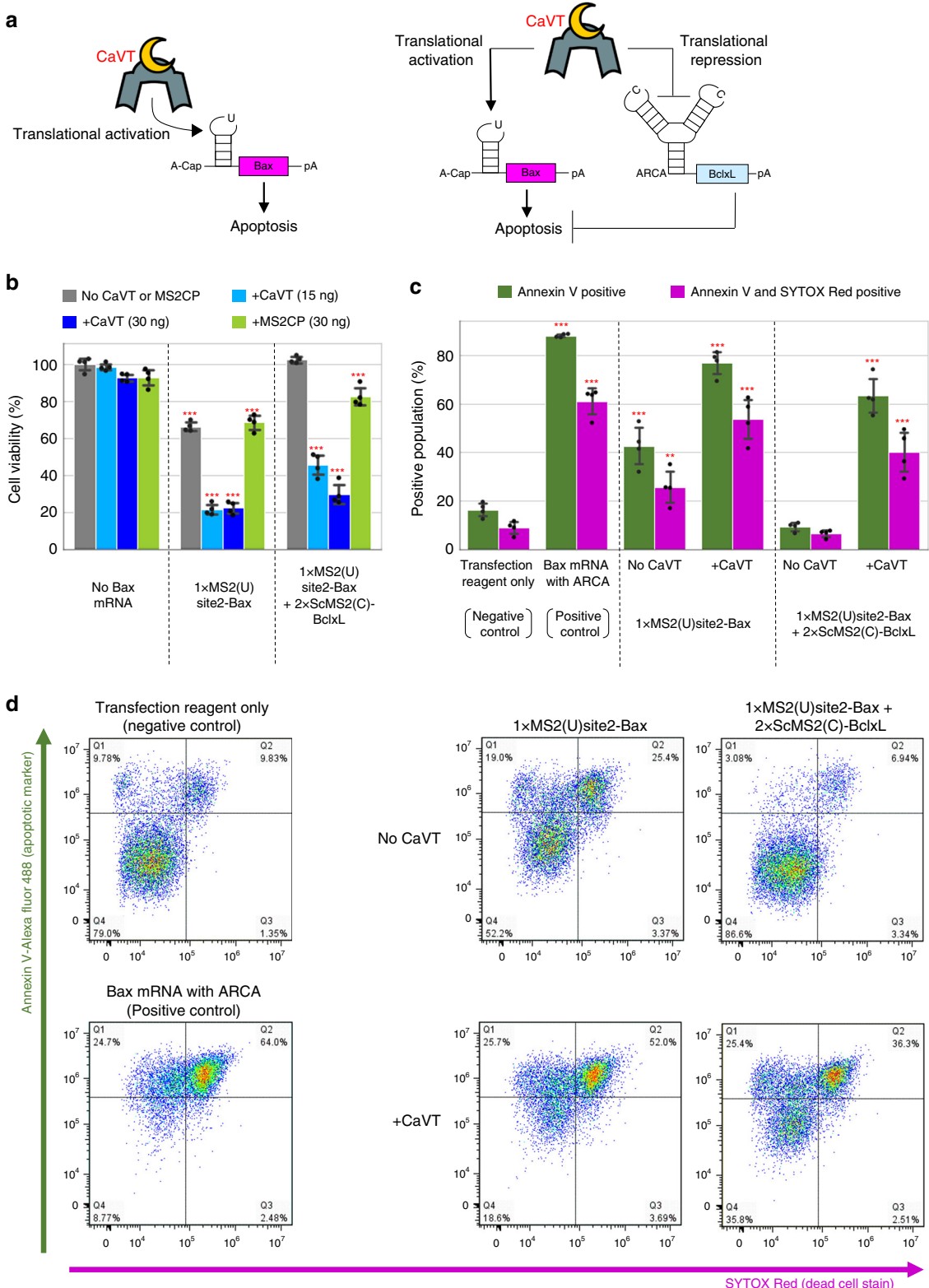

that contains a sequence complementary to its target DNA[23]. Cas9 is widely used for genome editing such as gene knockout or knock-in, because the cleavage site can be programmed by designing guide RNA sequences[24–26]. To check the CaVT-mediated translational activation of Cas9, we co-transfected 1xMS2(U)site2-SpCas9 mRNA and a single-guide RNA (sgRNA) that targets EGFP gene into EGFP-expressing HeLa cells (HeLa-EGFP) (Fig. 6a, top). As in the case of Bax, co-transfection of

CaVT increased the knockout of EGFP (approximately 60% of cells were EGFP negative), but some portion of EGFP knockout was observed in the absence of CaVT (approximately 40% of cells were EGFP negative), which is significantly higher than in non-treated cells (Fig. 6b, c). Therefore, we prepared mRNA encoding an anti-CRISPR protein, AcrIIA4, which binds with Cas9 to inhibit Cas9 DNA recognition and endonuclease activity[27–29]. The AcrIIA4-coding mRNA, named 2xScMS2(C)-AcrIIA4, has

**Fig. 5 Apoptosis regulation by CaVT-mediated translational activation and repression. a** Schematic diagrams of apoptosis-regulating circuits. Apoptosis is induced by only the translational activation of Bax (left) or both the translational activation of Bax and translational repression of Bcl-xL (right). **b** The viability of cells transfected with apoptosis-regulating circuits. HeLa cells were co-transfected with 1xMS2(U)site2-Bax (cap analog: A-cap), 2xScMS2(C)-BclxL (cap analog: ARCA), and CaVT or MS2CP mRNAs. All mRNAs contained N1mΨ. Cell viability was analyzed by the WST-1 assay. The bar graph shows the means ± SD. ($n = 4$ independent wells in each transfection condition). ***$P < 0.001$ compared to no Bax mRNA/no CaVT or MS2CP samples by ANOVA with Dunnett's multiple comparison test (two-sided). Exact $P$ values are shown in Supplementary Table 1. Source data are provided as a Source Data file. **c, d** Annexin V (apoptosis marker) and SYTOX Red (dead cell marker) staining. HeLa cells were co-transfected with 1xMS2(U)site2-Bax mRNA (cap analog: A-cap), 2xScMS2(C)-BclxL mRNA (cap analog: ARCA), and CaVT mRNA. For the positive control, 1xMS2(U)site2-Bax mRNA (cap analog: ARCA) was transfected. All mRNAs contained N1mΨ. One day after the transfection, the cells were stained and analyzed by a flow cytometer. The bar graph shows the average of four independent experiments (mean ± SD) (**c**). Representative two-dimensional dot plots (**d**). **$P < 0.01$, ***$P < 0.001$ compared to the negative control by ANOVA with Dunnett's multiple comparison test (two-sided). Exact $P$ values are shown in Supplementary Table 1. Source data are provided as a Source Data file.

the identical motif as 2xScMS2(C)-BclxL, which enables CaVT-mediated translational repression of AcrIIA4 (Fig. 6a, bottom). When co-transfected with CaVT mRNA, cells transfected with 1xMS2(U)site2-SpCas9 and 2xScMS2(C)-AcrIIA4 mRNAs showed high EGFP knockout efficiency comparable to the positive control, in which cells were transfected with Cas9 mRNA with translationally active cap analog (ARCA). On the other hand, in the absence of CaVT, the EGFP negative ratio of cells co-transfected with 1xMS2(U)site2-SpCas9 and 2xScMS2(C)-AcrIIA4 was comparable to that of non-treated cells (Fig. 6b, c). These results show that CaVT enables the efficient regulation of genome editing with simultaneous upregulation and down-regulation of Cas9 and AcrIIA4.

**Cell-selective regulation by miRNA-responsive CaVT.** We next investigated whether CaVT-based RNA circuits could detect endogenous signals and produce desired outputs in a cell-type-specific manner. We chose miRNAs as a representative marker, because there are various miRNAs and their activities depend on the cell type[30]. MiRNAs are small (about 22 nt) noncoding RNAs that regulate the translation of mRNAs through mRNA degradation or translational repression[31]. MiRNAs make complexes with Argonaute proteins (e.g., Ago2) and cleave or translationally repress mRNAs containing sequences partially or perfectly complementary to the miRNAs.

To achieve cellular state-dependent translational activation and repression in RNA circuits, we focused on miRNA-responsive mRNAs that we had previously used to sort or visualize specific cell types[21,26,32–34]. Thus, we designed CaVT mRNA that contains a complementary sequence to miR-21-5p or miR-302a-5p, two miRNAs highly expressed in HeLa and human iPS cells (hiPSCs, 201B7 strain), respectively. Because endogenous miR-302a-5p activity is very low in HeLa cells[26], when co-transfected with the apoptosis-inducing circuit composed of 1xMS2(U)site2-Bax and 2xScMS2(C)-BclxL (Fig. 7a) into HeLa cells, miR-302a-5p-responsive CaVT mRNA showed apoptosis induction that was comparable to conventional CaVT mRNA. The addition of miR-302a-5p mimic decreased cell death, which demonstrated the miRNA responsiveness of the system (Fig. 7b; Supplementary Figs. 8 and 9). To investigate whether the miRNA-responsive, apoptosis-inducing circuit can respond to endogenous miRNA, miR-21-5p-responsive CaVT mRNA was co-transfected with 1xMS2(U)site2-Bax and 2xScMS2(C)-BclxL into HeLa cells, in which endogenous miR-21-5p activity is high[26]. Different from conventional and miR-302a-5p-responsive CaVT, miR-21-5p-responsive CaVT did not induce apoptosis. The addition of miR-21-5p inhibitor restored the apoptosis induction, which indicated that the disappearance of apoptosis-induction was caused by the endogenous miR-21-5p-mediated repression of CaVT (Fig. 7b; Supplementary Figs. 8 and 9).

Next, we investigated whether miRNA-responsive CaVT can be used for cell-selective genome editing. For this purpose, EGFP-expressing stable cell lines of HeLa cells and hiPSCs were co-transfected with a genome editing-circuit composed of 1xMS2(U) site2-SpCas9, 2xScMS2(C)-AcrIIA4, EGFP-targeting sgRNA, and a miRNA-responsive or conventional CaVT mRNA (Fig. 7c). As in the case of the apoptosis-inducing circuit, in HeLa cells, EGFP-knockout induced by miR-302a-5p-responsive CaVT was as efficient as that by conventional CaVT. In contrast, the efficiency of miR-21-5p-responsive CaVT to induce EGFP-knockout was very low and at a level similar to non-treated cells (Fig. 7d top; Supplementary Fig. 10a). These results are consistent with the low miR-302a-5p activity and high miR-21-5p activity in HeLa cells. In hiPS cells, which have high miR-302a-5p and moderate miR-21-5p expression[26], the EGFP-knockout efficiencies induced by miR-302a-5p and miR-21-5p-responsive CaVT were approximately 1/4- and 1/2-fold that of conventional CaVT, respectively (Fig. 7d bottom; Supplementary Fig. 10b). These results indicated that CaVT can be a useful component for RNA circuits to achieve cell-selective regulation.

**Drug-inducible translational activation by split CaVT.** Drug-controllable gene-expression systems can modulate gene expressions at suitable levels and time windows. Drug-regulatable artificial transcriptional activators (e.g., tetracycline-inducible transactivator) have been widely used for DNA circuits, but they cannot be applied to RNA circuits. To construct a drug-regulatable CaVT applicable to conditional gene regulation in RNA circuits, we utilized the drug-responsive DmrA–DmrC hetero-dimerization system (a variant of the rapamycin-responsive FKBP–FRB hetero-dimerization system), in which DmrA and DmrC bind to each other in the presence of A/C heterodimerizer[35,36]. We fused MS2CP (or its V29I mutant) and VPg(FCV) with DmrA and DmrC, respectively, to make VPg (FCV) interact with target mRNAs only in the presence of the dimerizer (Fig. 8a). Then, we co-transfected twelve variants of target mRNAs (the six constructs shown in Fig. 3a with two modified nucleoside compositions (N1mΨ or Ψ/5mC)), MS2CP-1xDmrA mRNA, and DmrC-VPg(FCV) mRNA into HeLa cells, and incubated the cells in the presence or absence of the dimerizer. Translational activation of the target mRNAs was observed in all combinations, but the most suitable combination of nucleoside modifications and motif variants was N1mΨ-containing mRNA with a U-variant (Fig. 8b, c; Supplementary Figs. 11 and 12), which is consistent with the results of conventional CaVT (Fig. 3; Supplementary Fig. 4). However, different from the case of conventional CaVT, the most suitable location of the MS2-binding motif was site1 in the drug-regulatable CaVT. To confirm that the dimerizer-mediated translational activation was induced by an MS2CP-1xDmrA-DmrC-VPg(FCV) interaction-dependent mechanism, we also added the dimerizer

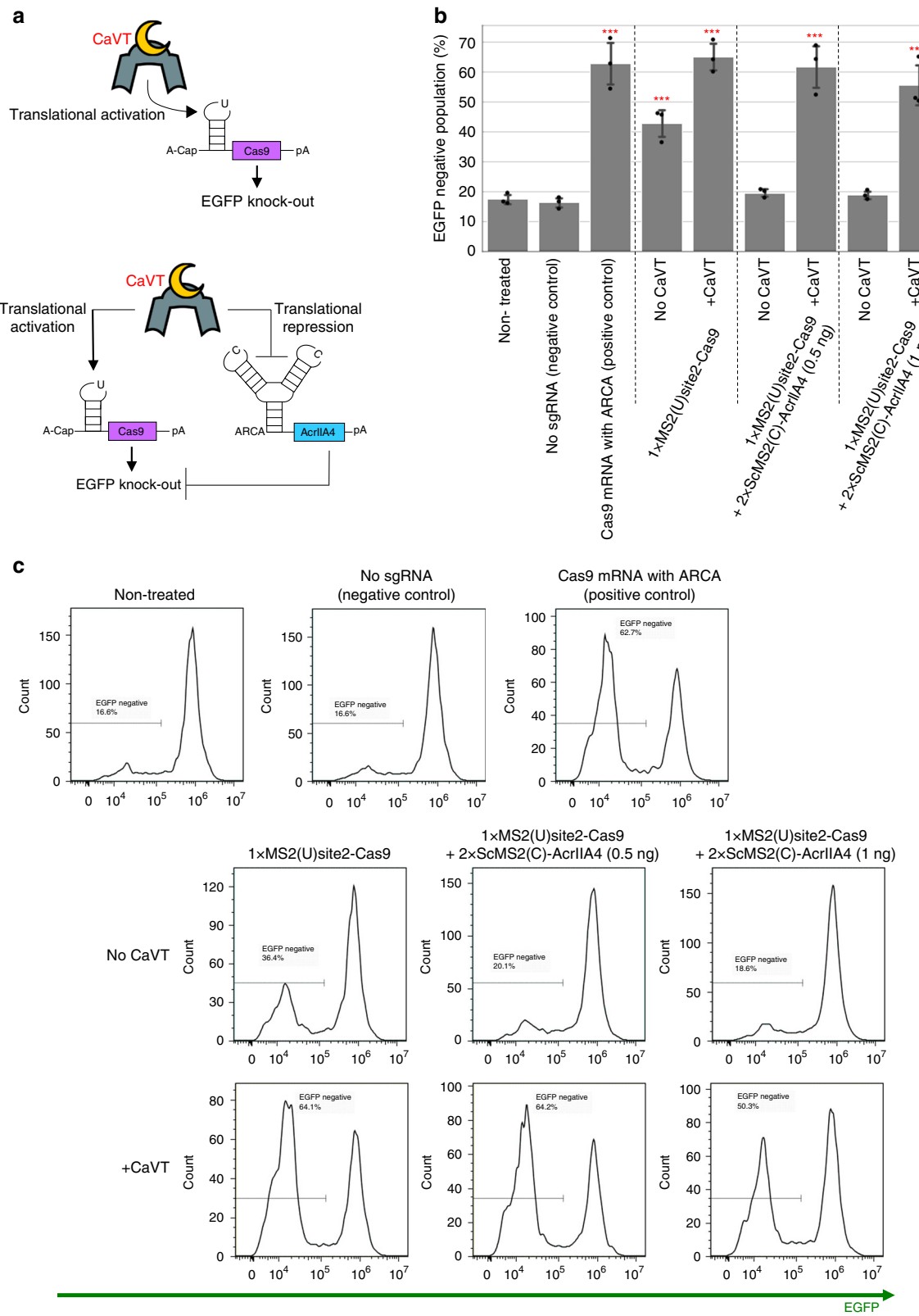

to cells without MS2CP-1xDmrA or DmrC-VPg(FCV). Dimerizer-mediated translational activation was observed only in the presence of both MS2CP-1xDmrA and DmrC-VPg(FCV) (Supplementary Fig. 13). To check the dose-dependency of the drug-regulatable CaVT, we incubated cells in a medium with various concentrations of the dimerizer after the co-transfection of 1xMS2(U)site1-hmAG1, MS2CP-1xDmrA, and DmrC-VPg

(FCV) mRNAs. As shown in Fig. 8d, e and Supplementary Fig. 14, a dose-dependent increase in the translation was observed. Finally, we captured time-lapse images to investigate the response time of the drug-regulatable CaVT. As early as 3 h after the dimerizer addition, an increase of hmAG1 fluorescence was observed (Fig. 8f; Supplementary Videos 1 and 2). Together, these results indicate that we could achieve conditional

**Fig. 6 Regulation of genome editing by CaVT-mediated translational activation and repression. a** Schematic diagrams of RNA circuits to regulate gene knockout. EGFP knockout is induced by only the translational activation of Cas9 (top) or both the translational activation of Cas9 and translational repression of AcrIIA4 (bottom). **b**, **c** The percentage of EGFP-negative cells transfected with RNA circuits to regulate EGFP knockout. HeLa-EGFP cells were co-transfected with 1xMS2(U)site2-SpCas9 mRNA (cap analog: A-cap), 2xScMS2(C)-AcrIIA4 mRNA (cap analog: ARCA), CaVT mRNA, and EGFP-targeting sgRNA. For positive and negative controls, 1xMS2(U)site2-SpCas9 mRNA (cap analog: ARCA) with or without EGFP-targeting sgRNA was transfected, respectively. All mRNAs contained N1mΨ. Five days after the transfection, EGFP fluorescence was analyzed by a flow cytometer. The bar graph shows the average of three independent experiments (mean ± SD) (**b**). Representative histograms (**c**). ***$P < 0.001$ compared to the non-treated samples by ANOVA with Dunnett's multiple comparison test (two-sided). Exact $P$ values are shown in Supplementary Table 1. Source data are provided as a Source Data file.

translational activation from synthetic mRNAs with our drug-regulatable CaVT system.

## Discussion

In the present study, we developed the novel translational activator, CaVT, which functions in RNA-based mammalian synthetic circuits. To date, several RBPs such as L7Ae, MS2CP, U1A, Lin28A, and TetR have been used to develop RNA circuits[9–11,37]. While the translational repression of mRNAs has been achieved by these RBPs, direct translational activation in mammalian cells is rather difficult. Thus, the "repression of translational repressors" approach has been used for indirect translational activation[9–11]. However, this layered approach makes the design of circuits complicated. In addition, the approach increases the components of the circuits, which increases the burden to cells. Therefore, the development of a method to directly activate mRNA translation is needed. We have previously developed L7Ae-mediated "ON switch", in which L7Ae-binding to mRNA prevents nonsense-mediated mRNA decay (NMD) to upregulate expression[38]. However, because nuclear pre-mRNA processing is necessary for NMD[39], mRNAs for NMD-based ON switches need to be transcribed in the nucleus. This limitation makes it difficult to use NMD-based ON switches in synthetic RNA circuits. In contrast, all components of our CaVT-mediated translational activation system can be transfected as synthetic mRNAs. Because synthetic mRNA does not cause insertional mutagenesis, it should be safer than DNA delivery, making it a promising method for medical applications such as gene therapies and regenerative medicines. Therefore, a translational activator compatible with RNA-only delivery will expand the use of synthetic gene circuits in clinical applications.

Different from FCV-derived VPg (VPg(FCV)), norovirus GI (NV-GI)-derived VPg (VPg(NV-GI)) showed high nonspecific translational activation (Fig. 2b–e; Supplementary Fig. 1). The MS2-independent translational activation may be caused by the direct binding of VPg to the G(5′)ppp(5′)A (a cap analog used to prepare A-capped mRNAs), because norovirus VPg can bind to nucleotide triphosphates directly[40]. FCV and norovirus VPg also differ in their interactions with translation-related endogenous proteins. For example, norovirus VPg directly interacts with eIF3, eIF4E, eIF4G, and polyA-binding protein (PABP), and its function is insensitive to eIF4E depletion[41–43]. In contrast, VPg(FCV) directly interacts with eIF4E, but not eIF4G or PABP, and it needs eIF4E-eIF4G interaction for translational activation[41,44]. Although VPg(NV-GI) showed higher translational activation than VPg(FCV) (Fig. 2b–e; Supplementary Fig. 1), we selected VPg(FCV) as a component of CaVT, because MS2CP-VPg(NV-GI) exhibited relatively high nonspecific translational activation (Fig. 2b–e). For example, the drug-inducible translational activation system (Fig. 8) requires that VPg only activates translation in an MS2 binding-dependent manner. In addition, norovirus VPg inhibits both Cap- and IRES-dependent translation[42], while VPg(FCV) does not affect that kind of translation[44]. Thus, VPg(FCV) seems to have less effect on non-target gene expressions

than VPg(NV-GI) and is thus more suitable to construct CaVT for mammalian gene circuits.

When we varied the insertion sites of the MS2-binding motif in the 5′ UTR, site2 (the center of the 5′ UTR) showed the highest translational fold change by CaVT (Fig. 3b; Supplementary Fig. 4a). The high fold change resulted from low basal expression in the absence of CaVT rather than high expression in the presence of it (Fig. 3c; Supplementary Fig. 4b). One possible explanation for the low basal expression of the site2-mRNAs is that the structure of the 5′ UTR contributes to VPg-independent translation. In the site1 and site3 constructs, the MS2-binding motif was simply added to the edge of the 5′ UTR. On the other hand, in the site2 construct, the MS2 binding motif interrupted the original 5′ UTR, which may affect the 5′ UTR structure and translation level.

Modified nucleosides such as N1mΨ, Ψ, and 5mC have been widely used for mRNA transfection to prevent immune responses and increase translations[3–5,45]. Recently, we found that such nucleoside modifications also affect the affinities of RBPs including MS2CP[18]. Because the binding of RBPs near the 5′ terminus of mRNAs decreases translation[16] and because RBP affinities to the mRNAs correlate with translational repression efficiency[46], we can estimate the relative affinity of MS2CP to each target mRNA with different nucleosides based on MS2CP-1xDmrA-mediated translational repression in the absence of A/C heterodimerizer (the condition in which DmrC-VPg(FCV) cannot interact with the target mRNAs, Fig. 8b). From the data of Fig. 8b (site1-mRNAs), we assume that the C-variant with N1mΨ has the highest affinity to MS2CP, followed by the U variant with Ψ/5mC, the U variant with N1mΨ, and the C variant with Ψ/5mC. Low-affinity combinations such as the U variant with N1mΨ and the C variant with Ψ/5mC showed higher translational activation (Fig. 3b; Supplementary Fig. 4a). One possible explanation for these observations is that the translational level is determined by the balance between VPg(FCV)-mediated activation and MS2CP-mediated repression, and that too strong binding to MS2CP could enhance the latter.

When we used site1-mRNAs as targets, CaVT V29I high-affinity mutant consistently showed lower translational activation than normal CaVT (Fig. 3b, c; Supplementary Fig. 4). As explained above, because the binding of RBPs such as MS2CP near the 5′ terminus of mRNAs repress translation[16], it is possible that translational repression by strong binding at site1 impairs VPg(FCV)-mediated translational activation. In most cases, CaVT and its V29I mutant showed similar effects on site2 and site3 mRNAs. However, in the case of C-variant with N1mΨ, the highest affinity motif, V29I mutant consistently showed lower translational activation than normal CaVT (Fig. 3b, c and Supplementary Fig. 4). These results suggested that strong binding can repress translation even when the binding sites are not near the 5′ terminus.

Translational activation by CaVT or MS2CP-VPg(NV-GI) became sluggish in cells with high transfection efficiency (i.e., cells with high tagRFP expression) when we used target mRNAs

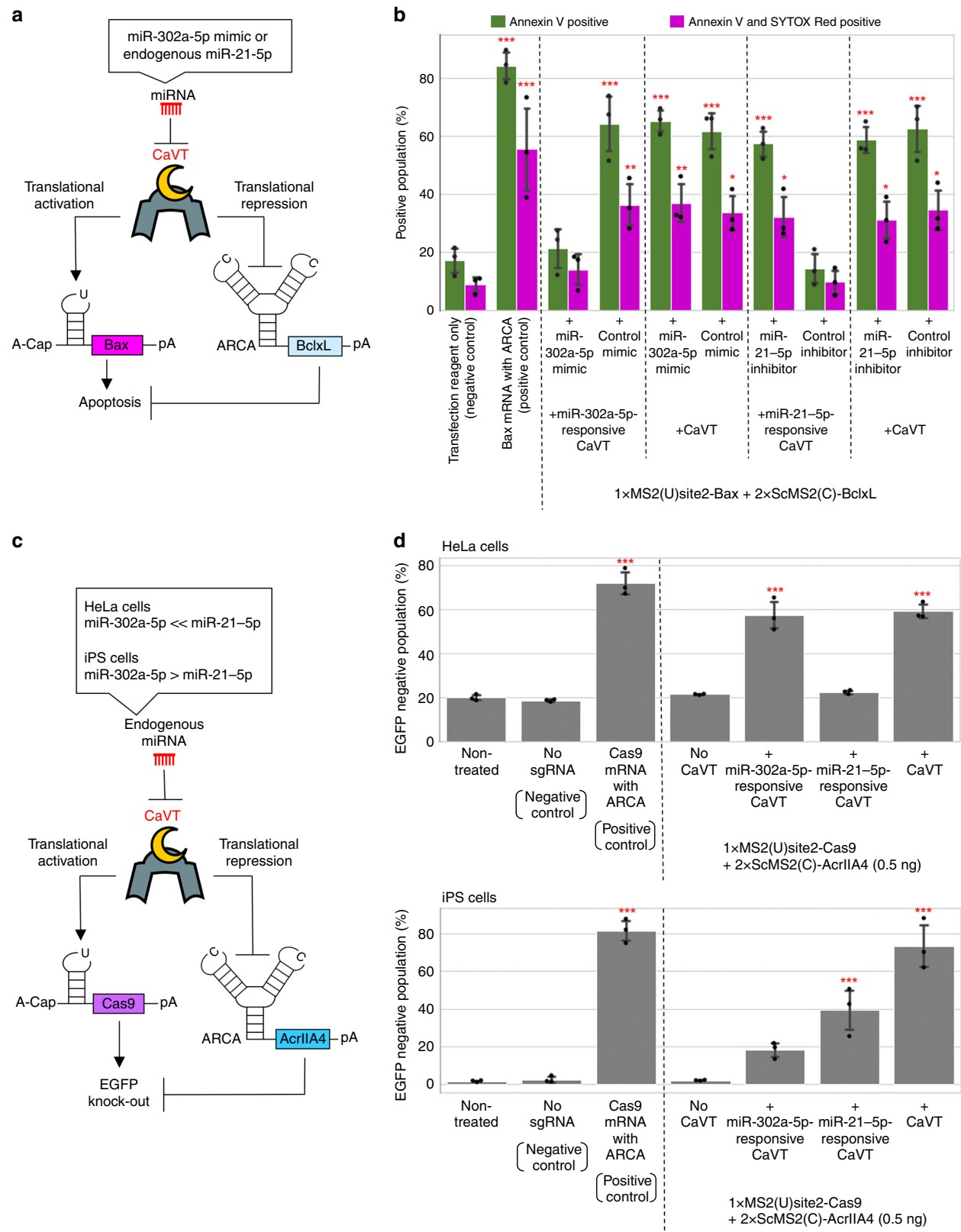

with an MS2 binding motif at site1 (Fig. 2d). This effect may be caused by a similar mechanism as the low translational activation of target mRNAs when using a high-affinity motif. Namely, if the concentration of CaVT or MS2CP-VPg(NV-GI) is high, these proteins may constantly bind target mRNAs to inhibit ribosome-scanning.

We also achieved drug-inducible translational activation by combining CaVT and drug-inducible hetero-dimerization domains (Fig. 8; Supplementary Figs. 11–14; Videos 1 and 2). Because inducible translation systems enable setting the expression level and time at suitable ranges for therapies, they will be an important component for future gene therapies. Although a

**Fig. 7 miRNA-mediated simultaneous regulation of translational activation and repression using miRNA-responsive CaVT mRNA. a, b** Apoptosis-inducing circuit regulated by miRNA. HeLa cells were co-transfected with 1xMS2(U)site2-Bax mRNA (cap analog: A-cap), 2xScMS2(C)-BclxL mRNA (cap analog: ARCA), the indicated CaVT mRNA, and the indicated miRNA mimic or inhibitor. For the positive control, 1xMS2(U)site2-Bax mRNA (cap analog: ARCA) was transfected. All mRNAs contained N1mΨ. One day after the transfection, the cells were stained with Annexin V and SYTOX Red followed by flow cytometry. A schematic diagram of the RNA circuit (**a**). The bar graph shows the average of three independent experiments (mean ± SD) (**b**). *$P <$ 0.05, **$P <$ 0.01, ***$P <$ 0.001 compared to the negative control by ANOVA with Dunnett's multiple comparison test (two-sided). Exact $P$ values are shown in Supplementary Table 1. Source data are provided as a Source Data file. **c, d** Cell-selective regulation of EGFP knockout by a miRNA-responsive genome editing circuit. The indicated EGFP-expressing stable cell lines were co-transfected with 1xMS2(U)site2-SpCas9 mRNA (cap analog: A-cap), 2xScMS2(C)-AcrIIA4 mRNA (cap analog: ARCA), the indicated CaVT mRNA, and EGFP-targeting sgRNA. For the positive and negative control, 1xMS2(U)site2-SpCas9 (cap analog: ARCA) with or without EGFP-targeting sgRNA was transfected, respectively. All mRNAs contained N1mΨ. Five days after the transfection, EGFP fluorescence was analyzed by a flow cytometer. A schematic diagram of the RNA circuit (**c**). The bar graph shows the average of three independent experiments (mean ± SD) (**d**). ***$P <$ 0.001 compared to the non-treated samples by ANOVA with Dunnett's multiple comparison test (two-sided). Exact $P$ values are shown in Supplementary Table 1. Source data are provided as a Source Data file.

drug-regulatable translational repressor was recently reported[11], to our knowledge, ours is the first report of a drug-regulatable translational activator for RNA circuits. Furthermore, because the domains of drug-regulatable CaVT are independent modules, the DmrA and DmrC used in this study can be exchanged with any other inducible hetero-dimerization domain[36,47]. Therefore, we can develop other regulatable CaVT according to the application. This feature is also suitable for the simultaneous use of multiple orthogonal CaVT regulated by different signals.

One important feature of CaVT is that the single protein enables different levels of translational activation and even translational repression by modulating the locations, sequences, and modified nucleosides of its binding motif (Figs. 3–4; Supplementary Figs. 3–7). Utilizing this feature, we demonstrated efficient regulation of apoptosis by the simultaneous translational activation of a pro-apoptotic protein and repression of an anti-apoptotic protein (Figs. 5 and 7a, b; Supplementary Figs. 8 and 9). Like apoptosis, there are many biological phenomena that are regulated by the balance of promoting and repressive factors. For example, cellular reprogramming and differentiation are regulated by the balance of transcription factors that contribute to the pluripotent and differentiated states[48]. One of our future goals is regulating complex cell-fate control by RNA circuits. Saxena et al.[7] reported a lineage control DNA circuit that inversely regulates the expression of multiple genes, including the simultaneous transcriptional activation and repression of Ngn3 and Pdx1, and succeeded to control hiPS cell differentiation into glucose-sensitive insulin-secreting beta-like cells. Although such lineage control has not been achieved with RNA circuits, the distinct feature of CaVT in terms of simultaneously upregulating and downregulating multiple genes seems suitable for this type of regulation.

The half-life of synthetic mRNAs is short, which is a suitable feature for applications that need only short-term transgene expression (e.g., purification of specific cells[21,33]), because the expression burden is low. However, some biological phenomena including reprogramming or differentiation need long-term gene expression. In such cases, the repeated incorporation of mRNAs into the cell is needed[49]. Another approach is the use of RNA replicons that are replicated and maintained long-term in mammalian cells[9,11,50]. While mRNAs expressed from RNA replicons have active 5′-cap structures, the target mRNAs of CaVT should not. Therefore, if we combine RNA replicons and CaVT, additional modifications to remove the 5′-cap may be needed. For example, the addition of self-cleaving ribozymes[51] at the 5′-ends may be helpful.

In summary, we have developed a novel translational regulator composed of an RBP and caliciviral VPg protein. This translational regulator, named CaVT, can be used for both translational activation and repression. Furthermore, it can be designed to exert miRNA-responsive, cell-selective gene regulation (Fig. 7; Supplementary Figs. 8–10). The cell-selective gene regulation by

CaVT may be used for the selective killing of cancer cells in cancer gene therapies or the elimination of harmful cells in cell transplantations. Additionally, we have developed drug-regulatable CaVT, which enables arbitrary gene regulation (Fig. 8; Supplementary Figs. 11–14). Finally, CaVT can be combined with many existing genetic tools including CRISPR–Cas9 and will help in the design of sophisticated mammalian gene circuits for biological studies and future medical applications.

## Methods

**pDNA construction.** KOD plus Neo (Toyobo, Osaka, Japan) or Q5 Hot Start High-Fidelity DNA polymerase (New England Biolabs Japan, Tokyo, Japan) was used for the polymerase chain reaction (PCR) to prepare the inserts. Oligo DNAs were purchased from Greiner Japan (Tokyo, Japan) or Eurofins Genomics K.K. (Tokyo, Japan). PCR and restriction digestion products were purified by the Monarch PCR & DNA Cleanup Kit (New England Biolabs Japan). All pDNAs were amplified in the E. coli strain DH5α or HST08 and purified using the PureYield Plasmid Miniprep Kit (Promega K.K., Tokyo, Japan). Details of the pDNA construction are described in the Supplementary Methods.

**In vitro transcription of mRNAs.** The template DNAs were prepared by PCR using Q5 Hot Start High-Fidelity DNA polymerase or PrimeSTAR MAX DNA polymerase (Takara Bio, Shiga, Japan). Oligo DNAs used for the PCR were purchased from Greiner Japan or Eurofins Genomics K.K. PCR products were purified by the Monarch PCR & DNA Cleanup Kit. Details of each template DNA are described in the Supplementary Methods. In vitro transcription was performed using the obtained template DNAs and a MEGAscript T7 Transcription Kit (Thermo Fisher Scientific K.K., Kanagawa, Japan). Totally, 6 mM Cap analogs (Anti Reverse Cap Analog (ARCA) (TriLink Biotechnologies, San Diego, USA) or G(5′)ppp(5′)A RNA Cap Structure Analog (A-cap) (New England Biolabs Japan)), 1.5 mM GTP, 7.5 mM ATP, CTP or 5-methyl-CTP, and pseudo-UTP or N1-methyl-pseudo-UTP (TriLink Biotechnologies) were used for the in vitro transcription. After the transcription, TURBO DNase (Thermo Fisher Scientific K.K.) was added to degrade the template DNAs. The obtained RNAs were purified by SPRI bead mix and dephosphorylated by rApid alkaline phosphatase (Roche Diagnostics K.K., Tokyo, Japan). The dephosphorylated RNAs were purified by using an RNeasy Minelute Cleanup Kit (Qiagen K.K., Tokyo, Japan). The length of the obtained RNAs was analyzed by MultiNA (SHIMADZU Corporation, Kyoto, Japan) or Agilent Bioanalyzer 2100 and an RNA 6000 Pico Kit (Agilent Technologies Japan Ltd., Tokyo, Japan).

**In vitro transcription of sgRNAs.** The template DNAs were prepared by PCR using Q5 Hot Start High-Fidelity DNA polymerase. Oligo DNAs used for the PCR were purchased from Greiner Japan or Eurofins Genomics K.K. PCR products were purified by the Monarch PCR & DNA Cleanup Kit. Details of each template DNA are described in the supplementary methods. In vitro transcription was performed using the obtained template DNAs and a MEGAshortscript T7 Transcription Kit (Thermo Fisher Scientific K.K.). All other steps were done as described in 'In vitro transcription of mRNAs' above.

**Cell culture.** HeLa cells (ATCC CCL-2) were maintained in Dulbecco's modified Eagle's medium (4.5 g/L Glucose, with L-Gln, without Sodium Pyruvate, liquid) (Nacalai Tesque, Kyoto, Japan) containing 9% fetal bovine serum, Antibiotic antimycotic solution (Sigma-Aldrich Japan, Tokyo, Japan), 0.09 mM MEM Non-Essential Amino Acids (Thermo Fisher Scientific), and 0.9 mM Sodium Pyruvate (Sigma-Aldrich Japan).

Human induced pluripotent stem cells (hiPS cells, 201B7-EGFP strain) were kindly provided by Dr. Knut Woltjen (Kyoto University). The hiPS cells were maintained in StemFit AK02N (Ajinomoto, Tokyo, Japan). The medium was

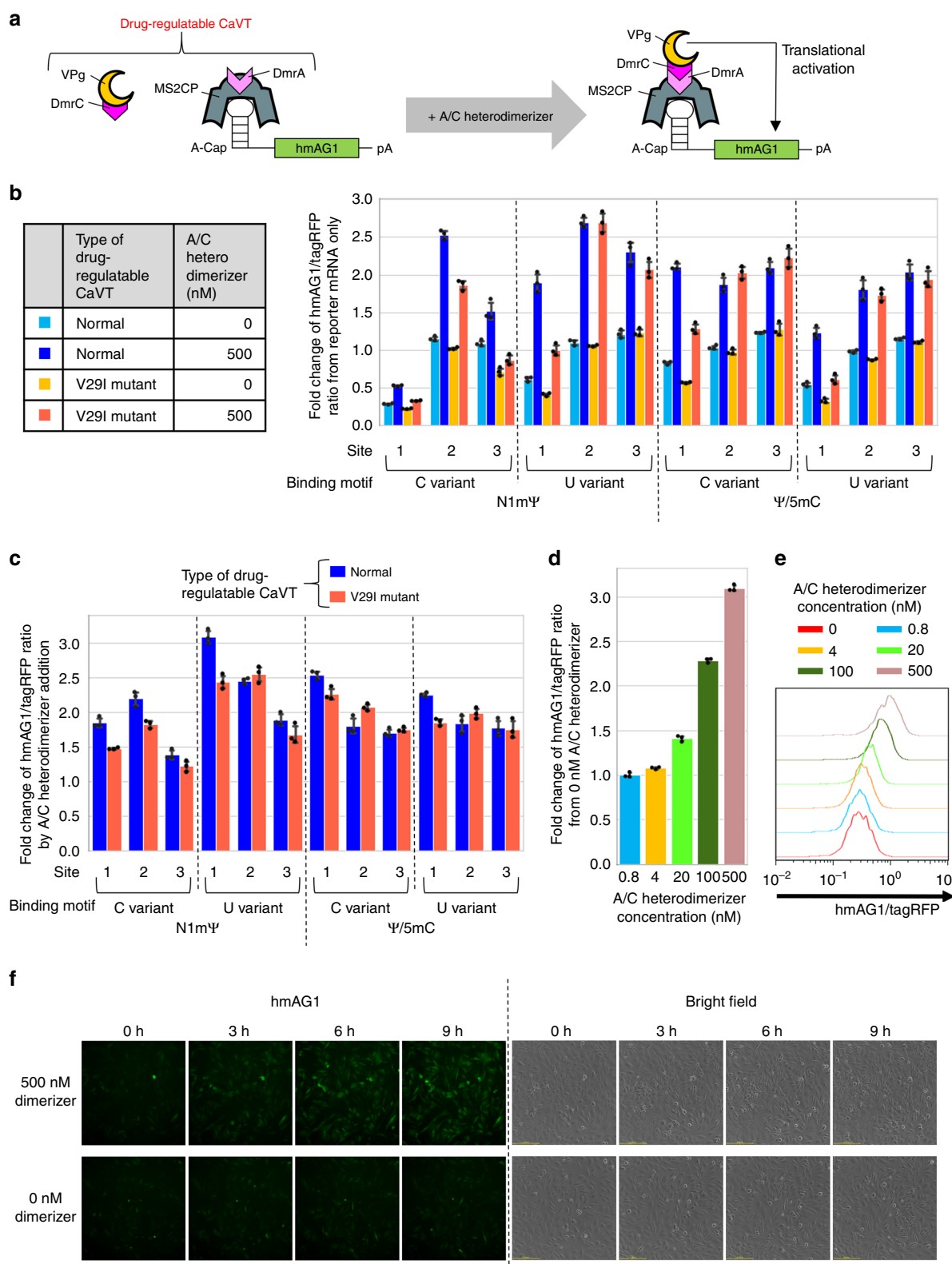

changed once every 2 days, and passage was performed once every 8 days using 0.5× TrypLE Select (Thermo Fisher Scientific) and cell scrapers. At the times of the cell seeding, plates were coated with iMatrix-511 silk (Nippi, Incorporated, ToKyo, Japan). More details are described in a previous report[52].

**Reporter mRNA transfection and flow cytometry.** Totally, $5 \times 10^4$ HeLa cells were seeded onto 24-well plates, and 1 day later they were transfected with the indicated amount of mRNAs using 1 μL/well of Lipofectamine MessengerMAX

(Thermo Fisher Scientific). One day after that, the cells were harvested using 200 μL/well of Trypsin/EDTA and suspended in 500 μL/well of DMEM. Then, the fluorescence was measured by a BD Accuri C6 with CFlow Plus software (BD Biosciences, San Jose, USA). The excitation wavelength was 488 nm for both tagRFP and hmAG1. Bandpass filters were 585/40 nm for tagRFP and 530/30 nm (90% attenuation) for hmAG1. Visualization of the data and calculation of the means of the hmAG1/tagRFP ratio in each cell expressing both hmAG1 and tagRFP were performed by FlowJo 7.6.5 (BD Biosciences). Microsoft Excel for Office 365 and Jupyter Notebook were used to depict bar graphs. The values of the fluorescence compensation (0.7–0.9% for 530/

**Fig. 8 Drug-mediated translational activation combining CaVT and the hetero-dimerization system.** HeLa cells were co-transfected with hmAG1 mRNAs containing an MS2-binding motif (cap analog: A-cap, modified nucleosides: N1mΨ or Ψ/5mC) and mRNAs that express tagRFP, DmrC-VPg(FCV), and MS2CP-1xDmrA or MS2CP(V29I)-1xDmrA. Then, A/C heterodimerizer was added. **a** Schematic diagram of drug-regulatable CaVT, which is composed of MS2CP-1xDmrA and DmrC-VPg(FCV). MS2CP-1xDmrA binds to the MS2 binding motif of target mRNAs. In the absence of A/C heterodimerizer, DmrC-VPg does not bind to DmrA, and there is no VPg-mediated translational activation. After the addition of A/C heterodimerizer, DmrA-DmrC interaction tethers VPg to the target mRNAs, and VPg activates translation. **b**, **c** Effects of modified nucleosides, sites, and variants of the MS2-binding motif and variants of MS2CP on translation. The fluorescence was measured by flow cytometry and means of the hmAG1/tagRFP ratio were normalized by the ratio in the reporter mRNA only (**b**) or 0 nM A/C heterodimerizer samples (**c**). The bar graph shows the average of three independent experiments (mean ± SD). Source data are provided as a Source Data file. **d**, **e** Dose-dependency of drug-regulatable CaVT. 1xMS2(U)site1-hmAG1 (cap analog: A-cap, modified nucleosides: N1mΨ), tagRFP, DmrC-VPg(FCV), and MS2CP-1xDmrA mRNAs were used for the transfection. Means of the hmAG1/tagRFP ratio were normalized by the ratio in 0 nM A/C heterodimerizer samples. The bar graph shows the average of three independent experiments (mean ± SD) (**d**). A representative histogram (**e**). Source data are provided as a Source Data file. **f** Time lapse images of drug-induced translational activation. 1xMS2(U)site1-hmAG1 (cap analog: A-cap, modified nucleosides: N1mΨ), DmrC-VPg(FCV), and MS2CP-1xDmrA mRNAs were used for the transfection. Each image was captured at the indicated time points after the addition of A/C heterodimerizer. Three independent experiments were performed to check reproducibility, and representative images are shown. The scale bar represents 200 μm.

30–585/40 nm and 36–38% for 585/40–530/30 nm) were determined based on the fluorescence of cells transfected with either hmAG1 or tagRFP. Details of the transfection conditions are shown in the Supplementary Methods.

**WST-1 assay.** Totally, $1 \times 10^4$ HeLa cells were seeded onto 96-well plates, and 1 day later they were transfected with the indicated amounts of mRNAs using 0.25 μL/well of Lipofectamine MessengerMAX. One day later, medium containing cell proliferation reagent WST-1 (Roche Diagnostics KK) was prepared (1:10 final dilution) and used to replace the medium of the transfected cells. After incubation for 1 h at 37 °C, absorbance wavelengths of 440 and 620 nm were measured by a microplate reader (Infinite M1000, Tecan Japan Co., Ltd., Kanagawa, Japan). Microsoft Excel for Office 365 and Jupyter Notebook were used to depict bar graphs. Statistical analysis was performed using R 3.6.1. Details of the transfection conditions are shown in the Supplementary Methods.

**Apoptosis assay by Annexin V and SYTOX red staining.** Totally, $5 \times 10^4$ HeLa cells were seeded onto 24-well plates, and 1 day later they were transfected with the indicated amounts of mRNAs using 1 μL/well of Lipofectamine MessengerMAX. One day later, the supernatants of each well were collected, and the cells were harvested using 200 μL/well of accutase (Nacalai Tesque). The harvested cells were suspended in the collected supernatants of each well and centrifuged at $200g$ for 5 min. The precipitated cells were stained with Annexin V, Alexa Fluor 488 conjugate (Thermo Fisher Scientific), and SYTOX Red (Thermo Fisher Scientific) for 30 min at room temperature. Then, 200 μL of PBS was added to each sample, and the fluorescence was measured by BD Accuri C6. The excitation wavelength was 488 nm for Alexa Fluor 488 conjugate and 640 nm for SYTOX Red. The bandpass filters were 533/30 nm for Alexa Fluor 488 conjugate and 675/25 nm for SYTOX Red. The obtained data were analyzed by FlowJo 7.6.5. Microsoft Excel for Office 365 and Jupyter Notebook were used to depict bar graphs. Statistical analysis was performed using R 3.6.1. Details of the transfection conditions are shown in the Supplementary Methods.

**EGFP knockout assay of HeLa cells.** Totally, $5 \times 10^4$ HeLa-EGFP cells[26] were seeded onto 24-well plates, and 1 day later they were transfected with the indicated amounts of mRNAs and an EGFP-targeting sgRNA using 1 μL/well of Lipofectamine MessengerMAX. One day later, the cells were passaged. Five days after the transfection, the fluorescence was measured by BD Accuri C6. The excitation wavelength and the bandpass filter were 488 and 533/30 nm, respectively. The obtained data were analyzed by FlowJo 7.6.5. Microsoft Excel for Office 365 and Jupyter Notebook were used to depict bar graphs. Statistical analysis was performed using R 3.6.1. Details of the transfection conditions are shown in the Supplementary Methods.

**EGFP knockout assay of iPS cells.** Totally, $5 \times 10^3$ 201B7-EGFP cells[26] were seeded onto 24-well plates, and 1 day later they were transfected with the indicated amounts of mRNAs and an EGFP-targeting sgRNA using 1 μL/well of Lipofectamine Stem (Thermo Fisher Scientific). The medium was changed once every 2 days. All other steps were done as described in 'EGFP knockout assay of HeLa cells' above.

**Time-lapse imaging.** Totally, $5 \times 10^4$ HeLa cells were seeded onto a 24-well plate, and 1 day later they were transfected with the indicated amounts of mRNAs using 1 μL/well of Lipofectamine MessengerMAX. Five hours after the transfection, the medium was removed, and 400 or 500 μL of fresh medium was added. Then the plate was moved into a Biostation CT (Nikon, Tokyo, Japan). Two hours later, 100 μL of medium containing 2500 nM A/C heterodimerizer (Takara Bio) was added to a well containing 400 μL of medium (final concentration = 500 nM). After the addition of A/C heterodimerizer, the time-lapse images were captured using the Biostation CT (objective lens: 10×) every 30 min. The filter set for fluorescence imaging was 472/520 nm. Exposure times were 4 and 400 ms for bright field and fluorescence imaging,

respectively. The contrast of all fluorescent images were identically adjusted by ImageJ[53]. Details of the transfection conditions are shown in the Supplementary Methods.

**Reporting summary.** Further information on research design is available in the Nature Research Reporting Summary linked to this article.

## Data availability
The authors declare that the data supporting the findings of this study are available within the paper and its Supplementary Information files. The source data underlying Figs. 2b, 3b, 4b, 5b, c, 6b, 7b, d, 8b–d, and Supplementary Figs. 1a, 4a, 7a, and 8 are provided as a Source Data file. If additional data are required, they are available from the corresponding author upon reasonable request.

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

## Acknowledgements

We are deeply grateful to Dr. Yoshihiko Fujita, Dr. Shunsuke Kawasaki, Ms. Karin Hayashi, Dr. Moe Hirosawa, Dr. Satoshi Matsuura, Mr. Sora Matsumoto, and Mr. Kaoru R. Komatsu (Kyoto University) for kind advice about the experimental procedures and conditions. We would like to thank Dr. Knut Woltjen and Dr. Akitsu Hotta (Kyoto University) for providing the 201B7-EGFP cells and genome editing-related vectors. We also thank Dr. Peter Karagiannis and Ms. Miho Nishimura (Kyoto University) for English proofreading and administrative support, respectively. This work was supported by the Kyoto University Education and Research Foundation, the Naito Foundation, and the Japan Society for the Promotion of Science JSPS KAKENHI [15H05722 to H.S. and 19K20696 to H.N.]. Some reagents were provided as prizes by NIPPON Genetics Co., Ltd.

## Author contributions

H.N. and H.S. devised the experimental design and wrote the manuscript. H.N. conceived the concept and carried out the experiments and data analysis.

## Competing interests

Kyoto University holds patents regarding microRNA-responsive systems (WO2015105172A1, WO2016010119A1, WO2016171235A1, WO2015141827A1, and US10378070B2). In addition, Kyoto University has filed a patent application regarding the translational regulators (JP2020015891). H.N. and H.S. are the inventors of record listed on the patents.
