## [Peer Review File · Nature Communications]

Reviewers' Comments:

Reviewer #1:

Remarks to the Author:

In their manuscript, Nakanishi and Saito present CaVET, the first drug-regulatable translational activator for RNA circuits. Because a 5' cap is not required for activity, this tool will be an asset to future findings in synthetic RNA circuitry, which has immense potential for applications in conditional gene regulation. The authors demonstrate the function of their activator by demonstrating its regulatory action in a number of experiments in mammalian cell culture, including measurements of apoptosis and fluorescence. All experiments are very well-designed and thorough, while the accompanying text explains the findings in excellent detail. The overall story could have been easily split into several publications. The supplementary information is also extensive—not only are the findings transparent, but they are presented so that they can be readily adopted by the field. There are a few minor suggestions to be addressed:

- Even though the figures show the intended action of the system pretty straightforwardly, it would be useful to have a summarizing schematic explaining all the outlined design principles and intended mechanisms to keep the overall intention of this work in mind.
- The language is very precise throughout, but there minimal typos to correct: some of the fold change graphs throughout the paper should likely say "binding" motif and line 244 should read "CRISPR."
- The "RNA secondary structure prediction" section of the Methods could be moved into the caption of Figure 2d.

Reviewer #2:

Remarks to the Author:

The manuscript by Nakanishi and Sato, entitled Caliciviral protein-based artificial translational activator for mammalian gene circuits with RNA-only delivery, provides a novel and exciting synthetic biology tool based on the properties of VPg. Vpg is the genome-linked viral protein driving protein synthesis in caliciviruses though genera-specific interactions with translation factors (eIF4E for FCV or eIF4G for MNV or HuNV). The authors have cleverly engineered gene expression based on this property by tethering VPg to reporter mRNA via the well-characterised MS2 system. First the authors tested different systems using MNV or FCV based VPg fusions, and while higher translational response is driven by the MNV VPg, the specificity measured is better for FCV VPg. The authors then characterised where best to position the RNA tether and whether optimising the MS2 binding motif improves the translatability of the targets. Surprisingly the authors discovered that more stable MS2 system resulted in reduced translatability of the system.

While the authors should do a better job at discussing the potential reason behind this (i.e. better tethering impact on re-initiation for several rounds of translation??) this provides them with a fully tunable translation system that can be tweaked to promote or reduce the translation of a gene target. Throughout the rest of the paper the authors apply the methodology to several examples showing that they can control cell fate by playing on the balance of apoptotic/antiapoptotic gene, that they can be used to regulate gene editing via the CRISPR/Cas editing by impacting on Cas9 expression and finally they demonstrate that the expression of the circuit can be regulated in a cell specific manner via miRNAs.

Overall, this is an elegant strategy that has been designed within the framework of synthetic biology and tunable translational enhancer/repressor. This clever idea makes the most of the unique properties of a viral protein and will be of use to other investigators in the field. Although the actual enhancing properties of the designed CaVET system are not very high, the selectivity can make it into a powerful. The experiments are well designed and controlled and as such I have no reservation in seeing this manuscript accepted without further work.

I only spotted a typo line 244 CRSPER which should read CRISPR.

Reviewer #3:

Remarks to the Author:

Summary

In this work, Nakanishi and Saito describe a new strategy called CaVET for control of gene expression in mammalian cells using translation initiation of mRNAs as a control point. CaVET consists of transfected 5'-MS2-tagged mRNA sequences that lack 5' caps along with a MS2 binding protein linked to a caliciviral cap analog protein called VPg. The binding interaction between the VPg-MS2BP and target mRNA is programmable and can simultaneously either activate or repress translation depending on the 5' UTR architecture and mRNA sequence. The authors use fluorescent protein expression from CaVET as a base case, then use it to specifically activate apoptosis and gene editing by relying on cell-specific miRNAs. In their last figure, they also build a chemical-inducible dimerization scheme for CaVET.

Significance

Overall, this work achieves an impressive scope of multiplexing the regulation of gene expression in mammalian cells using transfected mRNAs (which, as they point out, do not interfere with the genome) and miRNA-mediated selectivity of control is interesting. In particular, the authors present a very detailed study of variation in their design that could allow for tunable control. I think this work could be suitable for publication in Nature Communications, though I do have a few technical and pragmatic questions and concerns.

Major Criticisms

1. To what extent did the authors account for compensation/correction of bleed-through between green and red channels in their flow cytometry data? As far as I can tell, it isn't described in the Methods how this was done, and it would certainly impact the measured "fold activation" when normalized against the tagRFP. (A few histograms particularly in Figure 1d suggest that the hMAG1 isn't correlating linearly with tagRFP—which is surprising).

2. My biggest qualm with the work is that I really don't mechanistically understand the results in Figure 2. The claim that "the level of translational activation negatively correlates with the affinity in a certain range" isn't really represented by the data—the C vs. U trend is reversed between the N1M and the 5mc variants, and the CaVET mutants don't show universally lower ON signals either. It also opposes intuition that stronger affinity would decrease the translation initiation rate. The only trend that I'm really convinced by is that Site 2 variants give higher expression—but as described in the Discussion section, this seems likely to just be a sterics or RNA folding problem, not a question of RNA-aptamer affinity. Later figures replacing the fluorescent proteins with functional outputs suggest that these trends seem to be independent of the downstream transcript, which is sort of surprising. Overall, I would like to see some more discussion on how to further modulate this scheme for activation/repression, perhaps in a more rational and programmable way.

3. Given these contradictory results, I also wonder whether or not the mechanistic description can even be proven for translation activation—is it possible that the CaVET is just better protecting the mRNA target from nucleases? What data proves that this capping mechanism is actually happening effectively (considering the relatively moderate fold improvements relative to Reporter mRNA only, which I wouldn't expect to be translated at all?).

Minor Criticisms

4. A brief sentence should be added in the main text or methods section for how the CaVET itself was designed. The supplement suggests it's a translational fusion with an in-frame scar, was that optimized?

5. With respect to Figure 5, how are the concentrations of transfection RNAs determined? The 1:9/1:4 ratio is hinted at in the main text but is this a lever that you can turn to determine how effective is, e.g., a pro-apoptotic scheme vs. an anti-apoptotic one? I would like to see a little more description in the text or supplement where these numbers came from, or if there was any sort of optimization to get there?

6. In general, I find the description of Figures 6b/c confusing due to the all of the data normalizations and the color scheme shifting around (this is a pretty common thread throughout the data figures and it would be nice to be consistent). Can this be cleaned up?

7. The word "binding" is missing an N in Figure 6b.

8. Many of figures in the main body text are blurry and it would be best if the resolution can be improved. In particular, I cannot see the content of Figure 2d at all.

9. Line 244: typo on "CRISPR".

Dear Reviewers,

Thank you for your comments regarding our manuscript, “Caliciviral protein-based artificial translational activator for mammalian gene circuits with RNA-only delivery” . Here we submit our responses to those comments and attach a revised version of the manuscript.

However, we first would like to notify you about changes in the hmAG1/tagRFP ratios in the figures between the revised and previously submitted manuscripts. The changes were made because we found some bugs in the analytical software and re-analyzed the data. Although the absolute values were slightly changed, the overall tendency did not change. Therefore, the overall conclusion of the study was also unchanged.

In addition, for comprehensibility, we simplified the designation of the translational activator from “Caliciviral VPg-based Effector for Translation (CaVET)” to “Caliciviral VPg-based Translational activator (CaVT)” .

Our responses to all comments are below.

Point-by-point responses to the reviewers' comments

Reviewer #1:

General Comments:

In their manuscript, Nakanishi and Saito present CaVET, the first drug-regulatable translational activator for RNA circuits. Because a 5' cap is not required for activity, this tool will be an asset to future findings in synthetic RNA circuitry, which has immense potential for applications in conditional gene regulation. The authors demonstrate the function of their activator by demonstrating its regulatory action in a number of experiments in mammalian cell culture, including measurements of apoptosis and fluorescence. All experiments are very well-designed and thorough, while the accompanying text explains the findings in excellent detail. The overall story could have been easily split into several publications. The supplementary information is also extensive—not only are the findings transparent, but they are presented so that they can be readily adopted by the field. There are a few minor suggestions to be addressed:

Response:

We thank the reviewer for the positive feedback and address the suggested points below.

Comment 1:

– Even though the figures show the intended action of the system pretty straightforwardly, it would be useful to have a summarizing schematic explaining all the outlined design principles and intended mechanisms to keep the overall intention of this work in mind.

Response 1:

Thank you for the kind advice. As suggested, we added new Figure 1, which summarizes all the outlined design principles and intended mechanisms.

Comment 2:

– The language is very precise throughout, but there minimal typos to correct: some of the fold change graphs throughout the paper should likely say “binding” motif and line 244 should read “CRISPR.”

Response 2:

Thank you for the comment. We amended the words “Biding motif” in the figures and “CRSPER” in line 244 to “Binding motif” and “CRISPR”, respectively.

Comment 3:

- The “RNA secondary structure prediction” section of the Methods could be moved into the caption of Figure 2d.

Response 3:

Thank you for the advice. As suggested, we moved the description about the RNA secondary structure prediction from the Methods to the caption of new Figure 4a (previous Figure 2d).

Reviewer #2:

General Comments:

The manuscript by Nakanishi and Sato, entitled Caliciviral protein-based artificial translational activator for mammalian gene circuits with RNA-only delivery, provides a novel and exciting synthetic biology tool based on the properties of VPg. Vpg is the genome-linked viral protein driving protein synthesis in caliciviruses through genera-specific interactions with translation factors (eIF4E for FCV or eIF4G for MNV or HuNV). The authors have cleverly engineered gene expression based on this property by tethering VPg to reporter mRNA via the well-characterised MS2 system. First the authors tested different systems using MNV or FCV based VPg fusions, and while higher translational response is driven by the MNV VPg, the specificity measured is better for FCV VPg. The authors then characterised where best to position the RNA tether and whether optimising the MS2 binding motif improves the translatability of the targets. Surprisingly the authors discovered that more stable MS2 system resulted in reduced translatability of the system.

While the authors should do a better job at discussing the potential reason behind this (i.e. better tethering impact on re-initiation for several rounds of translation??) this provides them with a fully tunable translation system that can be tweaked to promote or reduce the translation of a gene target. Throughout the rest of the paper the authors apply the methodology to several examples showing that they can control cell fate by playing on the balance of apoptotic/antiapoptotic gene, that they can be used to regulate gene editing via the CRISPR/Cas editing by impacting on Cas9

expression and finally they demonstrate that the expression of the circuit can be regulated in a cell specific manner via miRNAs.

Overall, this is an elegant strategy that has been designed within the framework of synthetic biology and tunable translational enhancer/repressor. This clever idea makes the most of the unique properties of a viral protein and will be of use to other investigators in the field. Although the actual enhancing properties of the designed CaVET system are not very high, the selectivity can make it into a powerful. The experiments are well designed and controlled and as such I have no reservation in seeing this manuscript accepted without further work.

Response:

We thank the reviewer for the positive comment. Regarding the comment, we added some sentences that discuss the relationship between affinity and translational activation (Discussion section, Page 19, Line 5).

“Modified nucleosides such as N1mΨ, Ψ, and 5mC have been widely used for mRNA transfection to prevent immune responses and increase translations^{3, 4, 5, 44}. Recently, we found that such nucleoside modifications also affect the affinities of RBPs including MS2CP (Yi Kuang et al., in revision). Because the binding of RBPs near the 5' terminus of mRNAs decreases translation¹⁶ and because RBP affinities to the mRNAs correlate with translational repression efficiency⁴⁵, we can estimate the relative affinity of MS2CP to each target mRNA with different nucleosides based on MS2CP-1xDmrA-mediated translational repression in the absence of A/C heterodimerizer (the condition in which DmrC-VPg(FCV) cannot interact with the target mRNAs, Fig. 8b). From the data of Fig. 8b (site1-mRNAs), we assume that the C-variant with N1mΨ has the highest affinity to MS2CP, followed by the U variant with Ψ/5mC, the U variant with N1mΨ, and the C variant with Ψ/5mC. Low-affinity combinations such as the U variant with N1mΨ and the C variant with Ψ/5mC showed higher translational activation (Fig. 3b and Supplementary Fig. 4a). One possible explanation for these observations is that the translational level is determined by the balance between VPg(FCV)-mediated activation and MS2CP-mediated repression, and that too strong binding to MS2CP could enhance the latter.

When we used site1-mRNAs as targets, CaVT V29I high-affinity mutant consistently showed lower translational activation than normal CaVT (Fig. 3b-c and Supplementary Fig. 4). As explained above, because the binding of RBPs such as MS2CP near the 5'

terminus of mRNAs repress translation¹⁶, it is possible that translational repression by strong binding at site1 impairs VPg(FCV)-mediated translational activation. In most cases, CaVT and its V29I mutant showed similar effects on site2 and site3 mRNAs. However, in the case of C-variant with NImΨ, the highest affinity motif, V29I mutant consistently showed lower translational activation than normal CaVT (Fig. 3b-c and Supplementary Fig. 4). These results suggested that strong binding can repress translation even when the binding sites are not near the 5' terminus.”

Comment 1:

I only spotted a typo line 244 CRSPER which should read CRISPR.

Response 1:

Thank you for the comment. We amended the word “CRSPER” in line 244 to “CRISPR” .

Reviewer #3 (Remarks to the Author):

General Comments:

Summary

In this work, Nakanishi and Saito describe a new strategy called CaVET for control of gene expression in mammalian cells using translation initiation of mRNAs as a control point. CaVET consists of transfected 5' -MS2-tagged mRNA sequences that lack 5' caps along with a MS2 binding protein linked to a caliciviral cap analog protein called VPg. The binding interaction between the VPg-MS2BP and target mRNA is programmable and can simultaneously either activate or repress translation depending on the 5' UTR architecture and mRNA sequence. The authors use fluorescent protein expression from CaVET as a base case, then use it to specifically activate apoptosis and gene editing by relying on cell-specific miRNAs. In their last figure, they also build a chemical-inducible dimerization scheme for CaVET.

Significance

Overall, this work achieves an impressive scope of multiplexing the regulation of gene expression in mammalian cells using transfected mRNAs (which, as they point out, do not interfere with the genome) and miRNA-mediated selectivity of control is interesting. In particular, the authors present a very detailed study of variation in their design that could allow for tunable control. I think this work could be

suitable for publication in Nature Communications, though I do have a few technical and pragmatic questions and concerns.

Response:

We thank the reviewer for the positive and valuable comments, which we address below.

Major Criticisms

Comment 1:

1. To what extent did the authors account for compensation/correction of bleed-through between green and red channels in their flow cytometry data? As far as I can tell, it isn't described in the Methods how this was done, and it would certainly impact the measured "fold activation" when normalized against the tagRFP. (A few histograms particularly in Figure 1d suggest that the hmAG1 isn't correlating linearly with tagRFP—which is surprising).

Response 1:

Thank you for the important comment. In response, we added a new sentence describing the compensation procedure to the Methods section as shown below (Page 27, Line 5).

"The values of the fluorescence compensation (0.7-0.9% for 530/30-585/40 nm and 36-38% for 585/40-530/30 nm) were determined based on the fluorescence of cells transfected with either hmAG1 or tagRFP."

In addition, we added new sentences explaining why hmAG1 does not correlate linearly with tagRFP in some dot plots in new Figure 2d (previous Figure 1d) (Page 20, Line 14).

"Translational activation by CaVT or MS2CP-VPg(NV-GI) became sluggish in cells with high transfection efficiency (i.e., cells with high tagRFP expression) when we used target mRNAs with an MS2 binding motif at site1 (Fig. 2d). This effect may be caused by a similar mechanism as the low translational activation of target mRNAs when using a high-affinity motif. Namely, if the concentration of CaVT or MS2CP-VPg(NV-GI) is high, these proteins may constantly bind target mRNAs to inhibit ribosome-scanning."

Comment 2:

2. My biggest qualm with the work is that I really don't mechanistically understand

the results in Figure 2. The claim that “the level of translational activation negatively correlates with the affinity in a certain range” isn’t really represented by the data—the C vs. U trend is reversed between the N1M and the 5mC variants, and the CaVET mutants don’t show universally lower ON signals either. It also opposes intuition that stronger affinity would decrease the translation initiation rate. The only trend that I’m really convinced by is that Site 2 variants give higher expression—but as described in the Discussion section, this seems likely to just be a sterics or RNA folding problem, not a question of RNA-aptamer affinity. Later figures replacing the fluorescent proteins with functional outputs suggest that these trends seem to be independent of the downstream transcript, which is sort of surprising. Overall, I would like to see some more discussion on how to further modulate this scheme for activation/repression, perhaps in a more rational and programmable way.

Response 2:

Thank you for the important comment. We agree that the description in the previous manuscript was misleading. The description was written to indicate that the highest affinity motif is C variant with N1m Ψ and the lowest affinity motif is U variant with Ψ /5mC (i.e., [C variant with N1m Ψ] > [U variant with N1m Ψ], and [C variant with Ψ /5mC] > [U variant with Ψ /5mC]), which was incorrect. The actual order of affinities was newly estimated from new Figure 8b, which shows data of site1-mRNAs (previous Figure 6b). Our previous study (Endo K., et al., *Nucleic Acids Research*, vol. 41, 2013) indicated that the binding of MS2CP close to the 5’ terminus of mRNAs efficiently represses translation. Therefore, translational repression in 0 nM A/C heterodimerizer condition at site1 can be considered an indicator of the binding strength of MS2CP (in the absence of A/C heterodimerizer, there was only the effect of MS2CP binding and no effect of VPg on the target mRNAs). The actual affinity order estimated from this data (Fig. 8b, site1) is (C variant with N1m Ψ) > (U variant with Ψ /5mC) > (U variant with N1m Ψ) > (C variant with Ψ /5mC). This affinity order was fitted well with the results shown in new Figure 3b (previous Figure 2b) and new Supplementary Figure 4a (previous Supplementary Figure S3a).

As indicated, the CaVT V29I mutant showed a slight increase in translational activation at site2 and site3 of C variant with Ψ /5mC. These results can be interpreted as the binding between C variant with Ψ /5mC and normal MS2CP being lower than the optimum level, since the motif has the lowest affinity. New Figure 8b shows that the binding between U variant with N1m Ψ and normal MS2CP was comparable with

the binding between C variant with $\Psi/5mC$ and V29I mutant, and these combinations showed the highest translational activation in new Figure 3b and Supplementary Figure 4a.

In the case of site1, V29I mutant showed lower translational activation even when the motif was C variant with $\Psi/5mC$. As described above, binding to site1 induces translational repression (Figure 8b). Therefore, it is possible that enhanced affinity at this site causes a decrease in translational activation even in the case of C variant with $\Psi/5mC$.

Regarding this point, we modified the Results section (Page 8, Line 11).

Before revision

“Based on these results, we hypothesized that the level of translational activation negatively correlates with the affinity in a certain range. Thus, we designed hmAG1 mRNA containing two copies of the C-variant motif stabilized by a scaffold”

After revision

*“Based on these results, we hypothesized that **exceeding optimal affinity may decrease the translational activation level**. To examine this hypothesis, we designed hmAG1 mRNA containing two copies of the C-variant motif stabilized by a scaffold”*

In addition, we added several sentences to the Discussion (Page 19, Line 5).

“Modified nucleosides such as N1m Ψ , Ψ , and 5mC have been widely used for mRNA transfection to prevent immune responses and increase translations^{3, 4, 5, 44}. Recently, we found that such nucleoside modifications also affect the affinities of RBPs including MS2CP (Yi Kuang et al., in revision). Because the binding of RBPs near the 5' terminus of mRNAs decreases translation¹⁶ and because RBP affinities to the mRNAs correlate with translational repression efficiency⁴⁵, we can estimate the relative affinity of MS2CP to each target mRNA with different nucleosides based on MS2CP-1xDmrA-mediated translational repression in the absence of A/C heterodimerizer (the condition in which DmrC-VPg(FCV) cannot interact with the target mRNAs, Fig. 8b). From the data of Fig. 8b (site1-mRNAs), we assume that the C-variant with N1m Ψ has the highest affinity to MS2CP, followed by the U variant with $\Psi/5mC$, the U variant with N1m Ψ , and the C variant with $\Psi/5mC$. Low-affinity combinations such as the U

variant with N1mΨ and the C variant with Ψ/5mC showed higher translational activation (Fig. 3b and Supplementary Fig. 4a). One possible explanation for these observations is that the translational level is determined by the balance between VPg(FCV)-mediated activation and MS2CP-mediated repression, and that too strong binding to MS2CP could enhance the latter.

When we used site1-mRNAs as targets, CaVT V29I high-affinity mutant consistently showed lower translational activation than normal CaVT (Fig. 3b-c and Supplementary Fig. 4). As explained above, because the binding of RBPs such as MS2CP near the 5' terminus of mRNAs repress translation¹⁶, it is possible that translational repression by strong binding at site1 impairs VPg(FCV)-mediated translational activation. In most cases, CaVT and its V29I mutant showed similar effects on site2 and site3 mRNAs. However, in the case of C-variant with N1mΨ, the highest affinity motif, V29I mutant consistently showed lower translational activation than normal CaVT (Fig. 3b-c and Supplementary Fig. 4). These results suggested that strong binding can repress translation even when the binding sites are not near the 5' terminus.”

Comment 3:

3. Given these contradictory results, I also wonder whether or not the mechanistic description can even be proven for translation activation—is it possible that the CaVET is just better protecting the mRNA target from nucleases? What data proves that this capping mechanism is actually happening effectively (considering the relatively moderate fold improvements relative to Reporter mRNA only, which I wouldn't expect to be translated at all?).

Response 3:

Thank you for the important comment. Regarding this point, we performed new experiments and added new data (new Supplementary Fig. 2, shown below) that shows CaVT did not increase the expression from the mRNA capped with ARCA, a translationally active cap analog.

If the CaVT-mediated increase of expression was caused by an effect other than cap-mimicking (e.g., mRNA protection from nuclease), CaVT should also increase the expression of ARCA-capped mRNA. The dependency of the CaVT-mediated increase of expression on the absence of active caps indicates that the increase of expression is caused by a cap-mimicking effect.

Accordingly, we added several sentences about the new data (Page 6, Line 17).

*“To further confirm that the CaVT-mediated increase of translation was caused by a cap-mimicking effect of VPg, we investigated the effect of CaVT on 1xMS2(C)site1-hmAG1 mRNA already capped with translationally active cap analog (Anti-Reverse Cap Analog; ARCA). In the case of ARCA-capped 1xMS2(C)site1-hmAG1 mRNA, the binding of CaVT induced translational repression rather than translational activation (Supplementary Fig. 2), which was a similar effect as VPg-*unfused* MS2CP¹⁶.”*

Minor Criticism

Comment 4:

4. A brief sentence should be added in the main text or methods section for how the CaVET itself was designed. The supplement suggests it’s a translational fusion with an in-frame scar, was that optimized?

Response 4:

Thank you for the comment. When we designed the CaVT, we simply fused MS2CP and VPg. The indicated in-frame scar is a cutting site for the restriction enzyme (BamHI), which was necessary in the cloning procedure. Therefore, we did not perform any optimization for the in-frame scar sequence.

Comment 5:

5. With respect to Figure 5, how are the concentrations of transfection RNAs determined? The 1:9/1:4 ratio is hinted at in the main text but is this a lever that you can turn to determine how effective is, e.g., a pro-apoptotic scheme vs. an anti-apoptotic one? I would like to see a little more description in the text or supplement where these numbers came from, or if there was any sort of optimization to get there?

Response 5:

Thank you for the comment. To determine the optimum amount of RNAs for transfection, we performed several preliminary experiments. For example, in the case of the apoptosis assay, we tested 50~150 ng/well of pro-apoptotic gene mRNA in the 96-well plate condition and determined 90 ng/well as the best amount. For the 24-well plate condition, we simply increased the transfection amount 4-fold. Similarly, in the case of the EGFP knockout assay, we tested 0.5~80 ng/well of anti-CRISPR gene mRNA and determined 0.5-1 ng/well was optimal.

Comment 6:

6. In general, I find the description of Figures 6b/c confusing due to the all of the data normalizations and the color scheme shifting around (this is a pretty common thread throughout the data figures and it would be nice to be consistent). Can this be cleaned up?

Response 6:

Thank you for the comment. As suggested, we fitted the color scheme of new Figure 8b and c (previous Figure 6b and c). Figure 8b and c are based on identical data, and Figure 8c is shown to help readers easily grasp the effect of the dimerizer in each transfection condition. Therefore, in Figure 8c we cannot use the same normalization as in Figure 8b.

To be consistent with the color scheme, we also fitted the colors of bar graphs in new Figures 3b, 4b, 8d and Supplementary Figures 4a and 7a to the colors of the histograms.

Comment 7:

7. The word “binding” is missing an N in Figure 6b.

Response 7:

Thank you for the comment. We amended the word “Biding motif” in new Figure 8b (previous Figure 6b) to “Binding motif” .

Comment8:

8. Many of figures in the main body text are blurry and it would be best if the resolution can be improved. In particular, I cannot see the content of Figure 2d at all.

Response 8:

Thank you for the advice. As suggested, we improved the resolution of the figures.

Comment 9:

9. Line 244: typo on “CRISPR” .

Response 9:

Thank you for the comment. We amended the word “CRSPER” in line 244 to “CRISPR” .

Reviewers' Comments:

Reviewer #3:

Remarks to the Author:

the authors have addressed my concerns and I support publication.”.